# The Languini Kitchen: Enabling Language Modelling Research at Different Scales of Compute

## Abstract

The Languini Kitchen serves as both a research collective and codebase designed to empower researchers with limited computational resources to contribute meaningfully to the field of language modelling[1]. We introduce an experimental protocol that enables model comparisons based on equivalent compute, measured in accelerator hours. The number of tokens on which a model is trained is defined by the model's throughput and the chosen compute class. Notably, this approach avoids constraints on critical hyperparameters which affect total parameters or floating-point operations. For evaluation, we pre-process an existing large, diverse, and high-quality dataset of books that surpasses existing academic benchmarks in quality, diversity, and document length. On it, we compare methods based on their empirical scaling trends which are estimated through experiments at various levels of compute. This work also provides two baseline models: a feed-forward model derived from the GPT-2 architecture and a recurrent model in the form of a novel LSTM with ten-fold throughput. While the GPT baseline achieves better perplexity throughout all our levels of compute, our LSTM baseline exhibits a predictable and more favourable scaling law. This is due to the improved throughput and the need for fewer training tokens to achieve the same decrease in test perplexity. Extrapolating the scaling laws leds of both models results in an intersection at roughly 50,000 accelerator hours. We hope this work can serve as the foundation for meaningful and reproducible language modelling research.

## Contents

---

[1]See URL REDACTED

# 1   Introduction

Language modelling, a critical aspect of natural language processing (NLP), involves predicting the probability distribution over a sequence of words in a language. Its importance underpins a variety of NLP tasks such as machine translation (Vaswani et al., 2017), text generation (Brown et al., 2020), and question answering (Devlin et al., 2019). Presently, language modelling research primarily emphasises finetuning large pre-trained models (Ding et al., 2023; Zhang et al., 2023) as well as techniques on prompting (Liu et al., 2023) and programming with large language models (Schlag et al., 2023; Dohan et al., 2022) which have greatly improved performance across a variety of NLP tasks. However, this focus has inadvertently hampered the development of novel language modelling methodologies that require the model to be trained from scratch. The prevailing sentiment of "bigger equals better" can overshadow the potential benefits of alternative architectures and innovative methodologies, which may offer unique advantages.

Transformers, the backbone of this trend, have proven their efficacy by setting the standard across a broad spectrum of tasks (Vaswani et al., 2017). Interestingly, recent work shows how the Transformer can be derived from Fast Weight Programmers from the '90s (Schmidhuber, 1991; Katharopoulos et al., 2020; Schlag et al., 2021). However, Transformers are not without limitations. They exhibit issues such as quadratic computational complexity with respect to the sequence length, difficulty in capturing relevant tokens from large contexts (Tworkowski et al., 2023), and limitations due to the finite nature of its context (Dong et al., 2023). Furthermore, transformers have a large inference cost, which can pose a significant challenge when deploying models in resource-constrained environments (Chitty-Venkata et al., 2023; Bondarenko et al., 2021). These limitations underscore the need for continued refinement and innovation.

Additionally, recent work argues that published modifications to the vanilla Transformer architecture did not meaningfully improve performance on a question-answering task (Narang et al., 2021). After extensive empirical evaluation, the authors of that study conjecture that various improvements do not transfer across implementation and tasks — an issue also present in other machine learning areas such as e.g. recommendation systems (Ferrari Dacrema et al., 2019), optimisation (Sivaprasad et al., 2020; Choi et al., 2020), or generative adversarial networks (Lucic et al., 2018).

To address these challenges, we introduce the *Languini Kitchen*, a novel benchmark, codebase, and research collective. Languini Kitchen, or just Languini, aims to create an environment that enables researchers, in particular those with limited computational resources, to make meaningful contributions to language modelling research. This is achieved through an experimental protocol that constrains experiments to various scales of compute and through a public code repository that enables reproducible experiments. Languini is a blend of the words *language* and *linguine* where the latter is a type of pasta similar to spaghetti which ironically stands for the research nature of the code in the Languini code repository.

Recent work showed that the scaling laws are not universal across various model architectures (Tay et al., 2023). Furthermore, their results indicate that the vanilla transformer still comes out on top in a direct comparison with eleven other recently published models. To enable progress, Languini focuses on fair comparisons and reproducible results on complex and general benchmark. Different models are compared based on their performance trend as compute increases (Kaplan et al., 2020; Hoffmann et al., 2022) and the resulting scale plots will hopefully serve as a platform for identifying promising models or techniques that warrant further scale-up.

For evaluation, we use a filtered version of the books3 subset from The Pile (Gao et al., 2020) and the BigScience ROOTS corpus (Laurençon et al., 2022) which has been used previously as training data for various large language models (e.g. see Scao et al. (2022); Dey et al. (2023); Biderman et al. (2023); Touvron et al. (2023a;b)). After rigorous filtering, our version of the dataset consists of approximately 85GB of high-quality monolingual text from 158,577 published books which span a large variety of modern topics and stories that significantly surpass the complexity and size of previous academic benchmarks. Models which are trained on the Languini Books benchmark are compared at different compute scales based on their perplexity on held-out data. This includes out of distribution splits with books on certain topics (such as learning a new language) which are excluded from the training data in order to evaluate the model's predictive ability across several books as context.

Languini's open-source codebase provides a range of functions, from facilitating the model development process to logging mechanisms. The project is inspired by Scenic, a lightweight library that facilitates rapid prototyping of novel vision models (Dehghani et al., 2022). Similar to Scenic, the Languini codebase aims to keep the core functionality simple and prevents any dependencies between projects. Furthermore, researchers are encouraged to incorporate their projects into the Languini codebase, hopefully fostering a continually increasing collection of previous work to ease the comparison to new methods. In this work, we introduce the two initial Languini models: a feed-forward, GPT-based, decoder-only Transformer model (Section 4.2) and a recurrent quasi-LSTM (Section 4.3). For each model and each compute class, we empirically find the best hyperparameter configuration which results in scaling plots that allow the comparison of each model's scaling law.

In summary, our research contributions are the following:

- An experimental protocol for the comparison of language modelling research under different scales of compute.
- A high-quality filtering of the books3 datasets for language modelling research with out of distribution splits for evaluating long-range dependencies.
- A scaling law comparison between a GPT-based model and a quasi-LSTM model where the quasi-LSTM's scaling law is superior.
- A codebase for researchers to simplify development and enable fair and meaningful comparison with scalability in mind.
- An empirical analysis of byte-pair encoding tokenisation.

## 2 Background: Language Modelling

Language modelling is a central task in NLP where raw text is typically segmented into a sequence of words or subwords using a tokeniser, which operates based on a predefined vocabulary (Mikolov et al., 2010; Al-Rfou et al., 2019). These segmented units are commonly referred to as tokens. With this tokenised representation in place, the goal of a language model becomes the prediction of a subsequent token given its preceding sequence of tokens. This objective can be formally defined as maximising the probability of a sequence of tokens $w_1, w_2, ..., w_N$:

$$p(w_1, w_2, ..., w_N) = \prod_{t=1}^{N} p(w_t | w_0, ..., w_{t-1}) \tag{1}$$

where $p(w_t | w_0, ..., w_{t-1})$ is the probability of token $w_t$ given the sequence of previous tokens $w_0, ..., w_{t-1}$.

The performance of a language model can be evaluated using the total cross-entropy loss, which for a given dataset is defined as:

$$\mathcal{L} = -\sum_{t=1}^{N} \log p(w_t | w_0, ..., w_{t-1}) \tag{2}$$

The cross-entropy measures the negative log-likelihood of the observed data under the model. Lower loss indicates a better model, but comparing raw loss values can be unintuitive. Therefore, the loss is often transformed into perplexity, a more interpretable measure, defined as:

$$\text{PPL} = \exp\left(-\frac{1}{N}\sum_{t=1}^{N} \log p(w_t|w_0, w_1, ..., w_{t-1})\right) \tag{3}$$

$$= \exp\left(\frac{\mathcal{L}}{N}\right) \tag{4}$$

where the cross entropy, or average loss, is equal to $\frac{\mathcal{L}}{N}$, and $N$ is the number of tokens in the sequence.

Consider a model predicting from vocabulary of $M$ tokens that will predict $p(w_t|w_0, w_1, ..., w_{t-1}) = \frac{1}{M}$ for any $t$. Such a uniform model would have the following perplexity:

$$\text{PPL} = \exp\left(-\frac{1}{N}\sum_{t=1}^{N} \log p(w_t|w_0, w_1, ..., w_{t-1})\right) \tag{5}$$

$$= \exp\left(-\frac{1}{N}\sum_{t=1}^{N} \log \frac{1}{M}\right) \tag{6}$$

$$= \exp\left(-\log \frac{1}{M}\right) \tag{7}$$

$$= \exp\left(\log(M)\right) \tag{8}$$

$$= M \tag{9}$$

Thus, by exponentiating the average loss during training, perplexity can be interpreted as the effective vocabulary size of a uniform model.

While perplexity is a standard measure in language modelling, it has limitations. The perplexity measures of two methods are only directly comparable when the same tokenisation is used. This is because its value is influenced by the granularity of tokenisation which depends on the tokenisation algorithm and vocabulary size. A larger vocabulary increases the difficulty of each individual prediction (higher loss) but may reduce the number of predictions in total (lower $N$). Previous work has shown that this is not an equal trade-off as increasing the vocabulary has diminishing returns (see appendix in Hutchins et al. (2022)). Consequently, models trained with different tokenisers can produce perplexity values that are not directly comparable.

To alleviate this, we introduce the measure of normalised perplexity. This measure adjusts for differences in tokenisation granularity by dividing the cross-entropy with the total number of bytes of the decoded text $B$, rather than the number of tokens $N$:

$$\text{normalised PPL} = \exp\left(\frac{\mathcal{L}}{B}\right) \tag{10}$$

Normalised perplexity makes it possible to compare the same model trained on different tokenisers, as it provides a standardised measure that adjusts for the variability introduced by the choice of tokenisation. Naturally, if different models are compared using different tokenisation algorithms, it remains open if the relative difference is due to the choice of model or choice of tokenisation. Nevertheless, this measure ensures a more equitable comparison between methods and contributes to a more nuanced understanding of their relative performance. Furthermore, normalised perplexity is dataset independent, allowing also for a relative comparison of perplexity across different problems such as modelling natural language or modelling code.

## 2.1 Why Scalability Matters

The scalability of a language model refers to its ability to improve performance as more computational resources are invested, usually by training larger models on more training data. Scalability is a critical

aspect to consider when evaluating the potential of an architecture because it indicates how well the model can leverage additional resources.

Scaled-up language models, i.e. large language models (LLMs; see Zhao et al. (2023a) for a recent review), have demonstrated to be broad few-shot learners (Brown et al., 2020; Schulman et al., 2022; Chowdhery et al., 2022; OpenAI, 2023). LLMs excel on numerous tasks without or with little need for task-specific finetuning. They achieve excellent results on question answering, text summarisation, translation, and other NLP tasks (OpenAI, 2023), but are also increasingly applied to other modalities such as images (Saharia et al., 2022; Alayrac et al., 2022), audio Ghosal et al. (2023), and reinforcement learning settings Driess et al. (2023). However, raw LLMs do not align well with human values and additional work is necessary to transform a raw LLM into a robust, helpful, and harmless conversational agent (Bai et al., 2022a;b).

While the ability of LLMs on various downstream tasks can provide valuable insights, relying on downstream performance as the main measure for comparison presents several challenges. First, downstream performance keeps improving due to the development of new finetuning and prompting strategies (Hu et al., 2021; Liu et al., 2023). Thus, any fixed prompting strategy will quickly be outdated. Second, many evaluation datasets for LLMs are too difficult for models that were trained at smaller scales. Third, evaluating such datasets adds a considerable amount of complexity to the evaluation process. Lastly, downstream performance has been found to correlate strongly with pretraining perplexity (Raffel et al., 2020). For these reasons, in this work, we only focus on the perplexity on held-out data.

## 2.2 Existing Benchmarks

Language modelling has a rich history with a variety of benchmarks for model evaluation. Some notable examples include Penn Treebank (PTB, Mikolov et al. (2010)), WikiText-2 (WT2, Merity et al. (2017)), WikiText-103 (Merity et al., 2017), enwik8 and enwik9 Mahoney (2011), and Project Gutenberg (PG19, Rae et al. (2020)).

These datasets represent a broad range of sizes and complexity. The PTB and WT2 are tiny corpora with a limited vocabulary and little variety. The enwik8 and enwik9 datasets are used for evaluating the performance of compression algorithms. They consist of the first $10^8$ and $10^9$ bytes of an English Wikipedia XML dump from 2006. With just 1 GB of text, models often train multiple epochs on these datasets and are prone to overfit on the training data. WikiText-103 was created in 2016 and contains about 100M tokens from a fixed vocabulary of 103k different words resulting in about 515 MBs of data. It consists of preprocessed Wikipedia articles and has been often used as an academic language modelling benchmark since then. The issue with Wikipedia articles is their relatively short size. The average length of a Wikipedia article is about 3,600 words (approximately 4,300 tokens), limiting the length of long-term dependencies. The most recent and largest dataset is PG19. PG19 consists of 28,752 books with an average length of 69k tokens resulting in about 10 GB of data or about 2M training tokens when using a subword vocabulary of 32k. The PG19 dataset is large enough to train models with billions of parameters. However, all books were published over 100 years ago and thus don't reflect today's English language or diversity of topics.

Besides, on previous benchmarks models were often compared simply based on the average loss or perplexity on held-out data. While such comparisons offer insights, the best models are often also the most compute-intensive ones Brown et al. (2020). With the rise of well-funded industry labs, it has become increasingly difficult for academic labs to do research at that scale. E.g., all publications which advance the state of the art on PG19 are from Google or Google Deepmind with model sizes of up to 1.3B parameters (Hutchins et al., 2022). Training such models requires dedicated servers with multiple state of the art accelerators training for several days just to reproduce the results.

Recent work presents the idea of *cramming* experiments into a single day and a single consumer GPU (Geiping & Goldstein, 2023). In Section 3, we will also advocate for a shift away from unconstrained perplexity comparisons. While experiments offer valuable insights, they do not adequately account for the scalability factor, a key element in training large language models. The Languini benchmark, in an effort to demonstrate scalability, compares models based on different amounts of accelerator hours, resulting in a scaling plot or scaling law (Kaplan et al., 2020; Hoffmann et al., 2022). This approach seeks to provide a fair

and meaningful comparison of language modelling research at varying compute scales, thereby promoting inclusivity for research groups with limited funding resources.

## 3 The Languini Books Benchmark

The Languini Books benchmark represents a notable shift from previous language modelling benchmarks. It emphasizes reproducibility, scalability, and a comparison based on accelerator hours. By focusing on these aspects, Languini fosters a direct and effective comparison of different language models based on their performance at different scales of computational resources, aligning closely with the practical reality of training and evaluating such models.

### 3.1 Comparison based on Compute Class

A critical component of the Languini benchmark involves the concept of a *compute class*. This measure represents the number of accelerator hours (both parallel and sequential) spent during the training of the model. It diverges from the convention of comparing models based on their number of parameters or the total number of floating point operations (FLOPs).

The number of parameters or total FLOPs are hardware-agnostic metrics. However, these measures fall short of capturing the actual computational efficiency of the evaluated algorithms. Two models with an identical number of parameters or FLOPs can exhibit vastly different performances due to the model's underlying design and its ability to exploit the hardware's capabilities.

In particular, these hardware-agnostic metrics fail to account for the parallelisability of a model. As advancements in semiconductor technology, particularly in parallel computing and high-performance microarchitecture, continue to reshape the industry, models that scale well with an increased number of parallel processors can vastly outperform others given the same amount of total FLOPs.

On the Languini benchmark, the evaluation requires the measure of normalised perplexity (see Section 2) at different levels of accelerator hours spent. With accelerator hours increasing exponentially, this data serves to estimate the scaling law, helping researchers understand and extrapolate the trajectory of model performance as computational resources are scaled up further. In practice, the number of accelerator hours used in this paper is *not* the actual training time but is calculated before training based on a specific model's throughput (tokens per second) w.r.t. specific hardware. This increases flexibility as it allows the model to be trained on any hardware as long as the throughput is apriori measured w.r.t. the same reference hardware. The Languini codebase provides a script to measure the throughput of any PyTorch language model. Currently, this reference hardware is the Nvidia RTX 3090, chosen for its prevalence and accessibility in academic organisations and the compute classes considered in this work are 6, 12, 24, 48, and 96 hours. We use the following software versions PyTorch 2.0.0, Triton 2.0.0, Nvidia driver 535, and CUda version 12.2.

Consider an example where we have a specific model architecture with a given hyperparameter configuration (or *config* in short). We first evaluate its training throughput $v$ (number of tokens per second) on our reference hardware using an untrained instance with the throughput script provided by the Languini codebase. The throughput script uses the profiler of the DeepSpeed library (Rasley et al., 2020) to measure the time it takes to perform a forward pass, a backward pass, and a weight update for any PyTorch model. For a specific compute class given by $h$ accelerator hours, we can calculate the total number of tokens $T$ that we can process in that time: $T = 3600vh$. Given the total number of tokens $T$, we calculate the number of steps by dividing it by the number of tokens per batch which is batch size × sequence length × number of gradient accumulation steps.

Note that because we measured the throughput before training we do not actually need to train our model on our reference hardware. We can train on any other or even multiple accelerators as long as we use the same model config used to measure throughput. As we train the model we log the loss or normalised perplexity at certain training step intervals. To transform the learning curves from loss-over-steps into loss-over-accelerator-time we simply multiply the current step number by the number of tokens per step and

divide by the throughput of that model configuration on the reference hardware. This can be done during or after training.

Furthermore, it is also possible to approximately convert the compute class of $n$ hours on accelerator $A$ into $k$ hours on accelerator $B$ through the total number of tokens $T$. This is because given a model $M$ we can measure the throughput on the accelerators $A$ and $B$ and calculate the respective accelerator hours needed to consume the same number of tokens. E.g. training a specific GPT model for $T$ tokens takes 6h on an RTX 3090 but training the same config on an A100 takes 2.5h. We find that this factor is roughly constant throughout various scales of GPT models. Hence, future work may eventually move on to better hardware without the need to retrain all previous models. In Table 6 we included the accelerator hours for other common deep learning hardware that was available to us at the time.

A limitation of this method is that certain models might perform better on new hardware. As a result, the performance ratio between model X and model Y on hardware A might differ when tested on newer hardware B. Given the common use of GPUs to train models this is effectively already the case Hooker (2021). The use of a reference accelerator is mainly to enable effective compute constraints. Future researchers may decide to use different hardware for their evaluation. But for a fair comparison, previous work would have to be also evaluated on that reference hardware.

## 3.2   The Dataset

The Languini codebase is designed to support various datasets. In this work, we introduce the first dataset dubbed *Languini Books*. Languini Books is a filtered version from the popular books3 dataset, a subset of The Pile (Gao et al., 2020) which has been used as training data for various LLMs (e.g. Scao et al. (2022); Dey et al. (2023); Biderman et al. (2023); Touvron et al. (2023a;b)). The books3 dataset comprises a large collection of published books, encompassing approximately 101 GB of data.

We remove all books which are shorter than roughly 50 KB as they mostly consist of boilerplate text and little to no content. We also remove all non-English books as there are too few for any reasonable multilingual language modelling. To do so, we repeatedly sampled 200 bytes of text from each book and classify the language using langdetect (Joulin et al., 2016b;a) until we either sampled 50 times or one language has achieved above 90% presence. We then remove all books where English is not the most common language and with more than 5 non-English samples. The only exception here are books used for the language learning data split which we elaborate further in Section 3.2.1.

We tokenise all remaining books using a 32k SentencePiece model using BPE that was trained on the data of WikiText-103 (Merity et al., 2017). Through manual inspection, we find that books with relatively low average bytes per token are often undesired books with large amounts of numerical values (e.g. food calorie tables, price guides), non-latex mathematics, books with little natural text (e.g. a collection of artworks with titles, dates, author names, and auction prices, but obviously without images), or books with otherwise extensive unusual formatting (e.g. large number of lines for the reader to write down their own business plan). Upon manual inspection, we decided to remove all books with less than 3.2 average bytes per token.

Lastly, we train a Gensim Doc2Vec model (Rehurek & Sojka, 2011) to encode each book as a vector representation. We then use the cosine similarity measure to find exact and near duplicates. Previous work showed that even simple deduplication methods can speed up training significantly (Tirumala et al., 2023). After extensive manual inspection, we decided to remove any books that have a cosine similarity of 0.87 or higher. This captures various duplicates and near duplicates such as new editions or books which have been published again with slight differences (such as differences due to catering to British and American markets). This step resulted in the removal of 5,514 or 3.36% of books.

The final dataset consists of 84.5 GB of text data across 158,577 books with a total of 23.9B tokens given the WikiText-trained vocabulary. Each book has on average 559 KB of text or about 150k tokens, and a median of 476 KB of text or 128k tokens. We plot a T-SNE projection of the vector representations of the languini books in Figure 2 to visualise the diversity of the data. Furthermore, we distribute a list of filenames and a script with which the processed data can be extracted from the official books3 dataset.

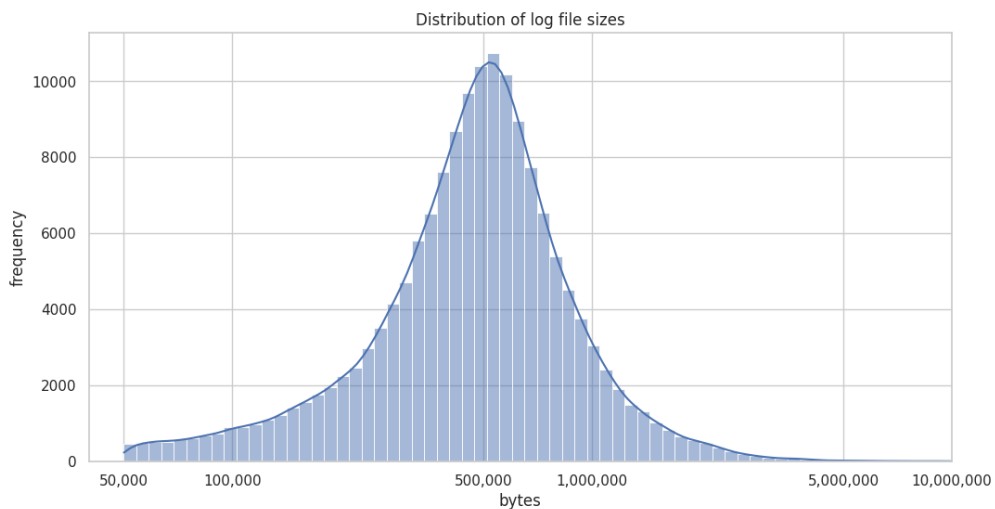

Figure 1: Distribution of book lengths (in bytes) of the Languini Books dataset.

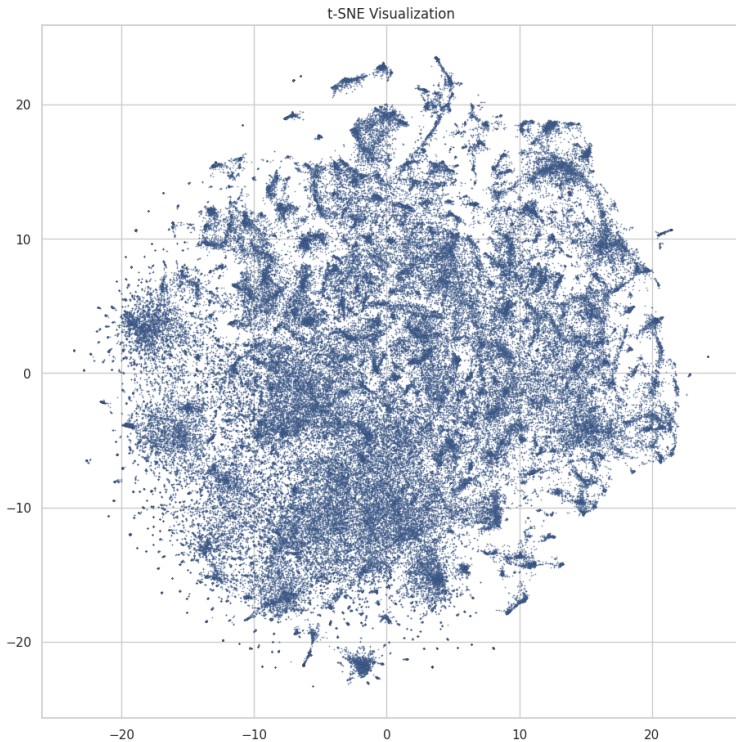

Figure 2: T-SNE plot of the learned vector representation for each book in the Languini Books dataset. Under a small Doc2Vec model, the books cluster into semantic concepts. The plot gives an impression of the large variety of books due to the large number of small clusters.

### 3.2.1 Evaluation and Test Sets

From the Languini Books data, we remove various books for evaluation purposes. This includes a standard i.i.d. test set with 80 books sampled at random. Furthermore, we create several out of distribution test sets to measure a model's ability to capture long dependencies and learn during inference through e.g. in-context learning (Dong et al., 2022; Kirsch et al., 2022), dynamic evaluation Krause et al. (2018), or meta-learning

(Irie et al., 2022; Kirsch & Schmidhuber, 2021). We split these test sets into the following categories: French Language Learning, Discworld, Java, Statistics, and Woodworking. The size of each of these sets is shown in Table 1.

| Split | Topic | Books | Bytes | Tokens | Bytes per Token |
|---|---|---|---|---|---|
| langlearn | French Language Learning | 34 | 16,571,748 | 6,582,737 | 2.52 |
| discworld | Discworld Series | 45 | 24,095,020 | 6,944,831 | 3.47 |
| java | Java Programming | 109 | 108,747,871 | 30,818,604 | 3.53 |
| stats | Statistics | 43 | 30,266,165 | 8,283,405 | 3.65 |
| wood | Woodworking | 19 | 7,846,725 | 2,146,089 | 3.67 |

Table 1: Size and topics of the books from every out of distribution split. Tokens and bytes per token are measured using WikiText tokenisation from Section 4.1.2.

**French Language Learning**  This dataset tests a model's ability to generalize to an unseen language under a curriculum. If the model is able to generalize online well, it should perform increasingly well on each sample in this dataset as it progresses through them. This ordered dataset consists of 17 French learning books with English text followed by 17 pure French books. Each subset is roughly ordered according to the perceived difficulty it would pose to a language model trained only on English text. As most books with non-English content were removed in the early preprocessing of the data, this dataset was constructed from a subset of the removed books. The French learning books were further selected for having a good balance of French and English tokens as well as having the word "French" in their title. The pure French books were arbitrarily taken from books that exclusively contained French tokens. Additional curation of the dataset was done heuristically. The dataset was ordered heuristically using titles and token counts as guides.

**Discworld**  The Discworld dataset consists of the available novels of Terry Pratchett set in the Discworld fantasy universe. There are 45 books in this dataset, with only 6 books missing from the main series. Books are ordered chronologically with the 35 available books in the main series first, then the 4 Science of Discworld books, and finally, the remaining 6 found books ordered arbitrarily. As the books have recurring characters, places, or themes, a language model able to generalize online well should perform increasingly well on each sample of the dataset and should do markedly worse on this dataset than it would on many other books in the principle datasets. As early processing of the dataset filtered similar books out of the dataset already, this dataset was constructed by searching for relevant keywords in the filenames of the books (i.e., "pratchett", "discworld", "diskworld", "josh kidby", or "paul kidby").

**Java**  This Java dataset focuses on the Java programming language. In total, there are 109 books in this dataset covering a wide variety of applications. There is no specific ordering for this dataset; however, a language model able to generalize online well should perform increasingly well, in expectation, as it processes subsequent samples of this dataset. This dataset was created by searching through all the books that contain both the string "java" and the string "public static" (the most common type declaration in java) anywhere in their full text. Through an analysis of the titles of the found books, books deemed to not be regarding Java or a library using Java were removed from all datasets.

**Statistics**  This dataset tests a model's ability to understand statistics while having little to no previous learning on the subject. It consists of 44 books. As with the Java dataset, there is no specific ordering to this dataset, but, in expectation, a model able to generalize online well should perform increasingly well as it processes subsequent samples of this dataset. This dataset was created by searching through the titles of books to find all that contained either the word "probability" or "statistics". An introductory textbook on statistics should most likely contain either of these terms. Subsequent filtering was done by hand.

**Woodworking**  This is a simple dataset consisting of only books related to woodworking projects. There are a total of 19 books in this dataset. There is no specific ordering to this dataset, but, as with the other datasets, a model able to generalize well online should, in expectation, perform increasingly well as it processes subsequent samples of this dataset. This dataset was created by searching for all book titles

containing "woodworking" or some variation of it. Project books with woodworking will mostly contain this word. Subsequent filtering was done by hand. Note that some of the books in this dataset use images to convey some information which is not available in the dataset.

## 4 The Baselines

In language modelling research, setting appropriate baselines is fundamental. Baselines offer a standard point of reference, facilitating the comparative analysis of new models or methodologies. For this study, we have selected two distinct architectures. The first is the widely-used GPT model, a highly parallelisable feed-forward architecture, for which we will conduct an in-depth performance analysis. The second is a recurrent architecture derived from the LSTM (Hochreiter & Schmidhuber, 1997). This section aims to detail these baseline models and their results. But before that, we will discuss tokenisation in Section 4.1.

### 4.1 Tokenisation Analysis

Tokenisation is a fundamental step in language modelling, aiming to convert input text into a sequence of tokens that a model can process. A common standard for tokenisation is currently the use of SentencePiece models (Kudo & Richardson, 2018) with Byte-Pair Encoding (BPE, Gage (1994); Sennrich et al. (2015)). These tokenisation models find the most frequent pairs of bytes in a text and repeatedly merge them to form tokens. Including all possible bytes as part of the initial vocabulary allows the tokeniser to encode any text and thus handle even words that have not been seen during training.

Existing LLMs utilise vocabularies of varying sizes. These range from thousands of tokens to over 200,000 tokens. Larger vocabularies are often used for large multi-lingual models such as GPT-3 (Brown et al., 2020), PaLM (Chowdhery et al., 2022), or Megatron-LM (Shoeybi et al., 2019). In this subsection, we delve into some of the challenges and intricacies of using a BPE tokeniser and analyse the performance implications of different vocabulary sizes.

#### 4.1.1 Analysing SentencePiece Vocabularies

In this subsection we present our analysis of various SentencePiece models which were trained on the training split of the Languini Books data. We analyse vocabularies from 2,048 to 131,072 unique tokens. The analysis reveals various shortcomings of the produced byte-pair encoding vocabularies which may be addressed in future work.

**SentencePiece constructs duplicate tokens that are never used.** In our experiments, we observed that SentencePiece tokeniser generates a number of tokens that have identical string decoding. For example, one such token is "b\xef\xbf\xbd" which is the UTF-8 encoding for the special character U+FFFD also known as the "replacement character". There are 128 different tokens that all decode to this character. Apart from this special character, there are 80 other tokens which we found to have duplicates mapping into the same string representation. These tokens typically represent single characters, such as lower- and upper-case letters, punctuation characters and special characters such as "#", "/", "_" etc. The two duplicate token groups are grouped together in terms of the order in the vocabulary. Interestingly, one group is at the "beginning" of the vocabulary, with token values below 200, whereas the other group is at the "end" of the vocabulary. Only one of these groups of tokens is actually used by the tokeniser to tokenise the text, ensuring that the tokenisation encoding-decoding process is deterministic. However, this phenomenon is wasteful, as in total there are 207 duplicate tokens that would ideally be used to encode other (sub-)words. We observed the same duplicate tokens across all vocabulary sizes we investigated and also across all data sets we used to extract these vocabularies.

**SentencePiece constructs near duplicates which make up 24.9% of the vocabulary.** Sentence-Piece by default encodes semantically identical words into different tokens, for example, we observed that "the", "The", " the", " The", "THE" and " THE" all get assigned to different tokens. See Table 2 for further examples. Making all token string representations lowercase and removing whitespace or punctuation marks

we found that there are 8,160 duplicate tokens in a vocabulary of 32,768. This does not include further similarities such as the genus, numerus, and kasus of different words. It may be counterproductive to have separate representations for such semantically similar strings because improving the token representation of one token does not translate to improvements in other tokens.

| subword | +space | +case | +space, +case | +caps | +space, +caps |
|---------|--------|-------|---------------|-------|----------------|
| "of" | " of" | "Of" | " Of" | "OF" | - |
| "ent" | " ent" | "Ent" | " Ent" | "ENT" | - |
| "was" | " was" | "Was" | " Was" | - | " WAS" |
| "not" | " not" | "Not" | " Not" | "NOT", | - |
| "from" | " from" | "From" | " From" | "FROM" | - |
| "house" | " house" | - | " House" | - | - |
| "love" | " love" | "Love" | " Love" | - | - |
| "chapter" | " chapter" | "Chapter" | " Chapter" | "CHAPTER" | " CHAPTER" |

Table 2: Further examples of vocabulary entries explained by simple transformations.

**A BPE vocabulary constructed from randomly sampled Languini Books contains 63% tokens which are identical with a vocabulary constructed from English Wikipedia.** When creating the Languini Books dataset (by filtering the books3 dataset) we constructed a BPE vocabulary of size 32,768 from the WikiText-103 dataset (Merity et al., 2017). We then constructed a BPE vocabulary of the same size by training on the Languini Books dataset. Due to the different nature of text contained in books compared to Wikipedia articles, we expected these vocabularies to have large differences, and the one trained on the Languini Books dataset to offer advantages for language modelling on this dataset. To our surprise, we found that the two vocabularies share 20,771 or 63% of all tokens.

**The frequency of the tokens follows approximately a Zipfian distribution.** Natural language has the property that the frequency of a word is approximately proportional to one over its rank when ordered (Zipf, 2013). We find that BPE vocabs closely follow this distribution. In Figure 3, we compare the frequency of tokens over their rank in the training data for vocabulary sizes ranging from 2,048 to 131,072. In this log-log plot, we normalised the rank of the token to the range $[0, 100]$ which means that a point on the x-axis should be interpreted as a percentage of all tokens since smaller vocabularies are rescaled to fit into the same plot with larger vocabularies. The sharp drop to the right is due to rare and unused tokens as discussed above. While it is probably advantageous that the learned vocabulary follows the same distribution as natural language text, this may also be a disadvantage because the number of training steps that are performed on a specific token is directly proportional to its frequency. We further note that a larger vocabulary follows the same trend and only reduces the frequency of each token. Thus, a large vocabulary may be undesirable because it "allocates" significantly fewer training steps for a significant number of tokens.

**Unigram tokenisation results in an almost identical frequency distribution.** In Figure 4 we compare vocabularies of size 32,768 encoded with either the BPE or unigram algorithm. The data suggests that unigram tokenisation results in a similar distribution except for the last 10% of the tokens where it appears to be more equally distributed compared to BPE. When trained, the language modelling performance between a unigram and BPE tokenisation with a vocabulary of 32,768 unique tokens resulted in no significant performance difference.

**The vocabulary distribution is uniform across the dataset.** In Figure 5, we compare the frequency distribution of 5 randomly sampled books with the distribution across 200 books. We find that the order of the tokens changes but the overall distribution is approximately identical.

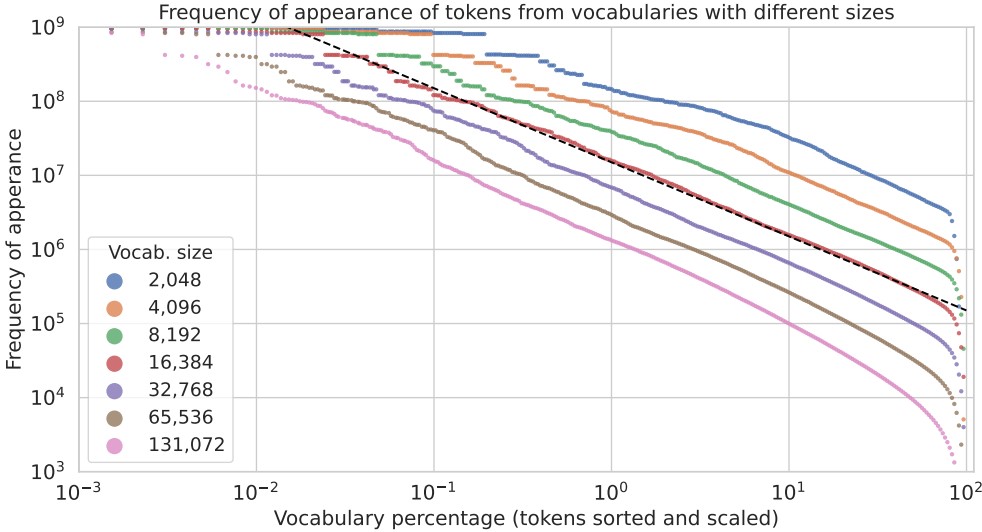

Figure 3: Token frequency sorted and scaled such that its rank (x-axis) lies within 0 and 100. The dashed line is the ideal Zipfian distribution scaled to fit the vocabulary with 16,384 unique tokens.

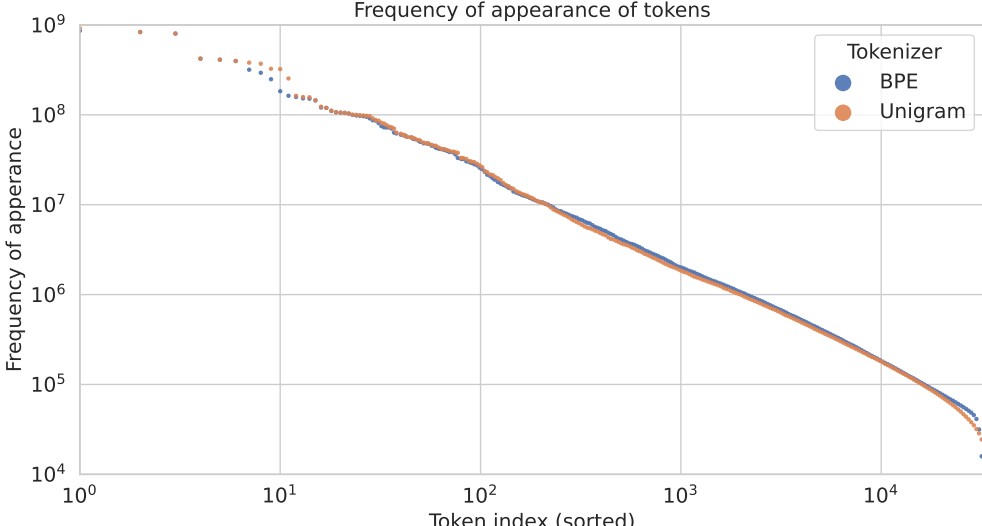

Figure 4: Comparison of the sorted token frequency for a vocabulary due to byte-pair and unigram tokenisation.

### 4.1.2 Performance Comparison of Different Vocabulary Sizes

Language models face the trade-off between tokenisation granularity and model complexity. A smaller vocabulary typically implies a higher granularity of tokenisation which in turn increases the total number of tokens to be processed. Conversely, a larger vocabulary results in a shorter training sequence as each token tends to represent more bytes.

A larger vocabulary is particularly useful for feed-forward models due to their finite context. A large vocabulary captures more bytes of raw text per tokens, thus the feed-forward model can condition on more text while processing the same number of tokens. However, larger vocabularies increase the number of parameters and reduce the throughput. For the purpose of Languini, the ideal vocabulary size strikes a balance between computational efficiency and model perplexity.

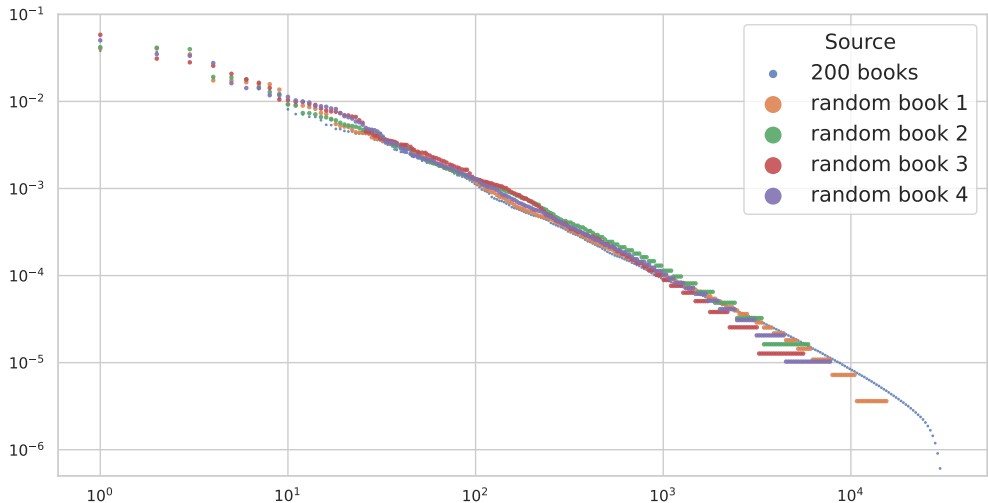

Figure 5: Comparison of the frequency distribution of 5 randomly sampled books against the distribution over 200 randomly chosen books.

In this section, we empirically explore the effects of different vocabulary sizes on normalised perplexity.

To study the impact of vocabulary size, we kept the model architecture (up to the embedding) and optimisation fixed and only change the granularity of tokenisation. We trained a GPT tiny model with a batch size of 160 for 138k steps across seven different vocabulary sizes from 2048 until 131,072. We present our results in Figure 6.

The findings from our comparison highlight a clear trade-off between vocabulary size and computational efficiency. Increasing the size of the vocabulary improves performance but has diminishing gains and significantly slow down the model (see Table 3). Our results indicate that a 16k and 32k vocabulary strikes a good balance. They offer improved performance without an excessive increase in computational costs. Since most models in languini will be on the smaller side, we decided on using a 16k vocabulary for the remaining experiments.

| vocabulary size | bytes per token | tokens per second |
|---|---|---|
| 2,048 | 2.84 | 183,065 |
| 4,096 | 3.21 | 174,568 |
| 8,192 | 3.54 | 163,448 |
| 16,384 | 3.81 | 141,434 |
| 32,768 | 4.02 | 112,855 |
| 65,536 | 4.17 | 78,778 |
| 131,072 | 4.25 | 45,742 |

Table 3: Best throughput of a GPT tiny model with different vocabulary sizes.

## 4.2   The Feed-Forward Baseline

With the decoder-only Transformer, the feed-forward approach has emerged as the new paradigm in language modelling. Its advantage is the processing and prediction of all sequence elements in a parallel manner. However, despite the theoretical ability to process longer sequences, the decoder-only Transformer model struggles to generalise to sequences that are significantly longer than what it has been trained with (Press et al., 2022). Although recent work has made progress in this regard (Luo et al., 2022).

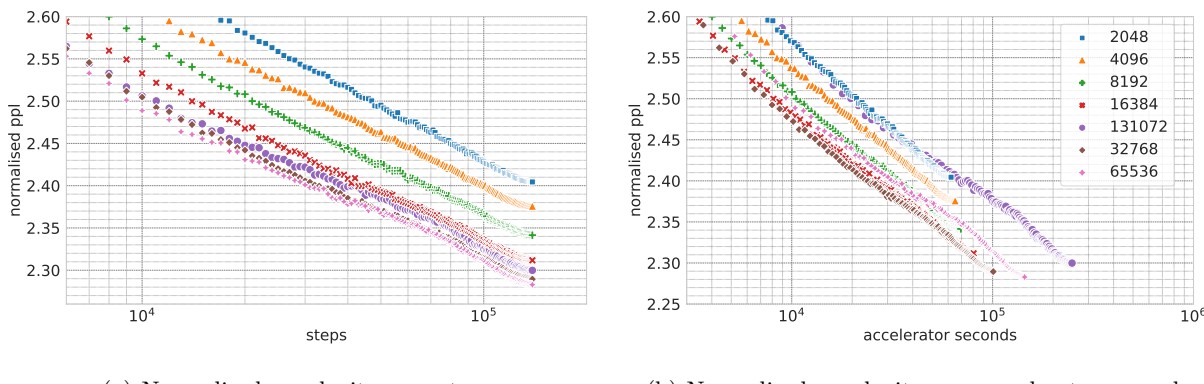

(a) Normalised perplexity over steps

(b) Normalised perplexity over accelerator seconds

Figure 6: Fast evaluation of normalised perplexity on held-out data for a GPT tiny model trained with a batch size of 160 for 138k steps across seven different SentencePiece BPE vocabulary sizes. Larger vocabularies require more memory and FLOPs to run which leads to different amounts of accelerator time.

The feature of the autoregressive decoder-only Transformer model to scale well with parallel compute allowed the scaling to hundreds of billions of parameters (Brown et al., 2020). Our implementation is based on the official TensorFlow implementation of GPT-2 (Radford et al., 2019; Karpathy, 2023).

Within our implementation, we provide configurations, i.e., hyperparameter settings, for different model sizes. Our configurations follow Radford et al. (2019) with ranges from 110M parameters to 1.5B parameters. To widen the scale of models to the lower end, we have included two additional sizes, namely, mini and tiny. All models are trained with a BPE vocabulary of 16,384 unique tokens as described in Section 4.1.

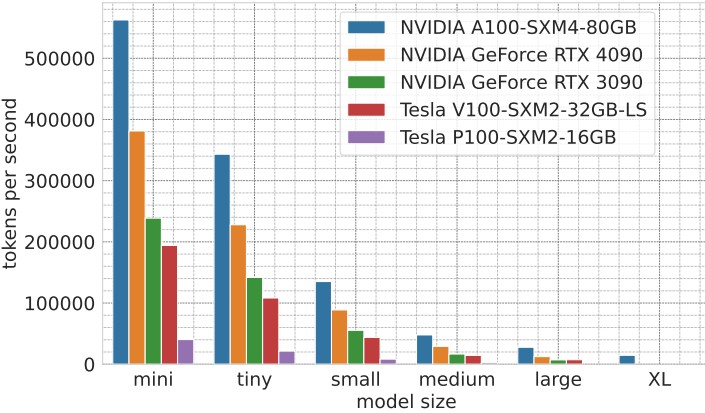

Figure 7: Maximum throughput (tokens per second) achieved on various accelerators for all GPT model sizes.

The decoder-only transformer architecture consists of multiple transformer layers. Each layer consists of two sublayers: The self-attention and position-wise multi-layer perception (MLP). The self-attention mechanism allows the model to weigh the significance of the proceeding tokens in the input sequence relative to the current token. Using multi-head attention, it can capture various types of relationships in the data. The position-wise MLP sublayer is applied next. It consists of 2 linear maps with the GELU non-linearity applied in between **??**. Typically, the middle activations of the MLP are four times the size of the hidden representation. Both sublayers are connected residually (Srivastava et al., 2015; Hochreiter, 1991) and are preceded by layer normalisation (Ba et al., 2016). We add a learned position embedding to each token before the first layer.

| GPT Model | gigaFLOPs | Params | $d_{\text{model}}$ | $n_{\text{layers}}$ | $n_{\text{heads}}$ | $d_{\text{head}}$ |
|-----------|-----------|--------|--------|---------|---------|--------|
| mini | 20.4 | 27.6M | 512 | 4 | 8 | 32 |
| tiny | 45.1 | 53.9M | 768 | 4 | 12 | 64 |
| small | 109.6 | 110.6M | 768 | 12 | 12 | 64 |
| medium | 352.4 | 336.4M | 1024 | 12 | 16 | 64 |
| large | 760.5 | 731.1M | 1536 | 24 | 16 | 96 |
| XL | 1,555.2 | 1,478.2M | 2048 | 24 | 24 | 128 |

Table 4: Overview of our GPT model sizes, flops, parameters, and differing hyperparameters. Sizes are chosen such that they are comparable in size and flops with the GPT models. FLOPs are the total number of floating point operations during the forward pass with a batch size of 1 as measured by the DeepSpeed flops profiler.

Our implementation leverages PyTorch's support for mixed precision training (Huang et al., 2020). In this mode, suitable operations (e.g. matrix multiplication) are executed with float16, leveraging tensor cores, while parameters for which more precision are kept in float32 which is crucial. The conversions are done automatically. This not only enhances training speed but also ensures stability. Additionally, we leverage the compile functionality of PyTorch 2.0.0 and Triton 2.0.0 (Wu, 2023) to further improve a model's throughput.

All models are trained with the Adam optimiser and with a cosine learning rate decay schedule which decays the starting learning rate of 0.0006 by a factor of 100 over the total number of training steps.

It's worth mentioning that our experiments did not incorporate the recently published flash attention mechanism (Dao et al., 2022), despite its growing popularity. In our experiments, the use of flash attention results in significant speedups but coincides with unexpected and large gradient spikes when used on our reference hardware (Nvidia's RTX 3090), hindering the model's ability to recover and continue training. In contrast, our native PyTorch implementation displayed stability throughout the entire training process of all experiments. The issue seemingly lies with certain hardware requirements of flash attention (Dao, 2023). We leave it for future work to provide an in-depth experimental evaluation of flash attention within the constraints of the Languini Books benchmark.

To calculate the affordable number of training steps of every GPT model for various compute classes, we measured their throughput w.r.t. our reference hardware. For future work, we also documented the throughput of all our GPT models on other accelerators here and in the GPT project folder of the GitHub repository.

| GPT Model | RTX 3090 | RTX 4090 | P100-16GB | V100-32GB | A100-80GB |
|-----------|----------|----------|-----------|-----------|-----------|
| mini | 238,719 (97) | 381,174 (96) | 40,292 (56) | 194,130 (320) | 562,594 (320) |
| tiny | 141,864 (76) | 227,890 (64) | 21,452 (40) | 108,221 (104) | 343,182 (272) |
| small | 55,416 (34) | 88,701 (32) | 8,310 (16) | 43,839 (50) | 135,107 (128) |
| medium | 16,618 (10) | 29,188 (11) | 2,503 (3) | 14,387 (17) | 47,851 (53) |
| large | 7,058 (4) | 12,379 (5) | OOM | 7,296 (9) | 27,713 (36) |
| XL | OOM | OOM | OOM | OOM | 14,559 (20) |

Table 5: Overview of the best tokens per second measures and their respective batch sizes in brackets for different GPT model sizes on several accelerators. OOM stands for out of memory.

### 4.2.1 Evaluation

We evaluate the GPT models using a fast and slow evaluation. To track performance (average loss, normalised perplexity, or others) during training, we evaluate the models on the held-out data for just 500 batches, using a batch size of 16, the default sequence length of 512 tokens, and, most crucially, by averaging the loss across all predictions. This is a low estimate of the actual performance of the model because the context for every prediction varies from 1 to 512 tokens and predictions with less context perform significantly worse (Kaplan et al., 2020). Ideally, we would measure performance with a batch size of 1 and such that each token has the

largest possible context. However, due to the nature of our implementation and the use of learned position embeddings, this would require us to recompute all previous tokens for every prediction.

We choose a middle ground where we evaluate over the last 128 tokens using a batch size of 1. We refer to this evaluation as the slow evaluation because it requires the processing of four times as many batches. In practice, We found that the normalised perplexity of our models does not significantly improve when evaluating over the last 64 or the last 32 tokens in the sequence but are exponentially more expensive. Thus, the slow evaluation of all our GPT models is only evaluating the last 128 predictions of the sequence which results in a context length between 384 and 512 tokens. Consequently, during evaluation, we slide our batches in the temporal direction not by the sequence length of 512 tokens but by 128 tokens which requires us to process four times as many batches.

### 4.2.2 Results

Given a specific model size, its ideal throughput, and compute class, we calculate the total number of tokens we can process in that time. In Figure 8, we present the fast evaluation results (i.e. loss averaged over all tokens of a batch instead of only the over the last $n$ predictions) for the trade-off between batch size and number of training steps for all model sizes. Note that larger models have slower throughputs which means that they process fewer tokens given the same compute class.

Our results indicate that the compute-optimal batch size for all GPT models lies roughly between 256 and 512 elements. All models are trained with a sequence length of 512 which results in the optimal number of tokens per batch to be roughly between 131k and 262k. As seen in Figure 8, the compute-optimal batch size seems to increase slightly as the same size model is trained for longer. This may indicate that the compute-optimal batch size is not constant throughout training but increases as previous work suggests (Smith et al., 2017; McCandlish et al., 2018; Shallue et al., 2018; Zhang et al., 2019). This insight is also supported by the common strategy of increasing the batch size towards the end of a long training run when training large language models. We leave it for future work to propose strategies for adaptive batch sizes and how they relate to the common practice of decaying the learning rate.

In Table 6 we summarise the normalised perplexity results for each compute class by evaluating the last 128 tokens of each batch as elaborated in Section 4.2.1 (unlike the results in Figure 6 and 8 which plot the fast evaluation). Note, that the hours on our reference hardware can be converted into hours on any other hardware through the total number of tokens (see Section 3.1 for further details). Using the normalised perplexity at different scales, we create a scale plot over accelerator seconds and FLOPs in Figure 9. As expected, perplexity over accelerator time closely follows a power law.

| compute class | normalised perplexity | config | batch size | total train tokens | total exaFLOPs | *theoretical A100 hours | *theoretical V100 hours |
|---|---|---|---|---|---|---|---|
| 6h | 2.262 | small | 128 | 1.2B | 0.769 | 2.46 | 7.58 |
| 12h | 2.197 | small | 128 | 2.4B | 1.538 | 4.92 | 15.17 |
| 24h | 2.146 | small | 256 | 4.8B | 3.075 | 9.84 | 30.34 |
| 48h | 2.087 | medium | 256 | 2.9B | 5.930 | 16.67 | 55.44 |
| 96h | 2.032 | medium | 256 | 5.7B | 11.859 | 33.34 | 110.89 |

Table 6: Average normalised perplexity results evaluated with a context of 384 or more of the overall best GPT runs of each compute class (based on the slow eval). The hours in each compute class are w.r.t. the ideal RTX 3090 throughput as measured by the Languini throughput script. From the hours and throughput, we calculate the total number of tokens to be processed during training. The total number of floating point operations is calculated by (total train tokens/sequence length) × forward flops of the model × 3. As in previous work, we estimate the FLOP count of the backward pass to be double the forward pass (Kaplan et al., 2020). *Given the total number of tokens we can compute the ideal accelerator hours for other hardware based on the throughput of the same model config.

We also evaluate the best GPT models for each scale on the out of distribution splits. The results are listed in Table 11 below together with the evaluation of the qLSTM model of Section 4.3.

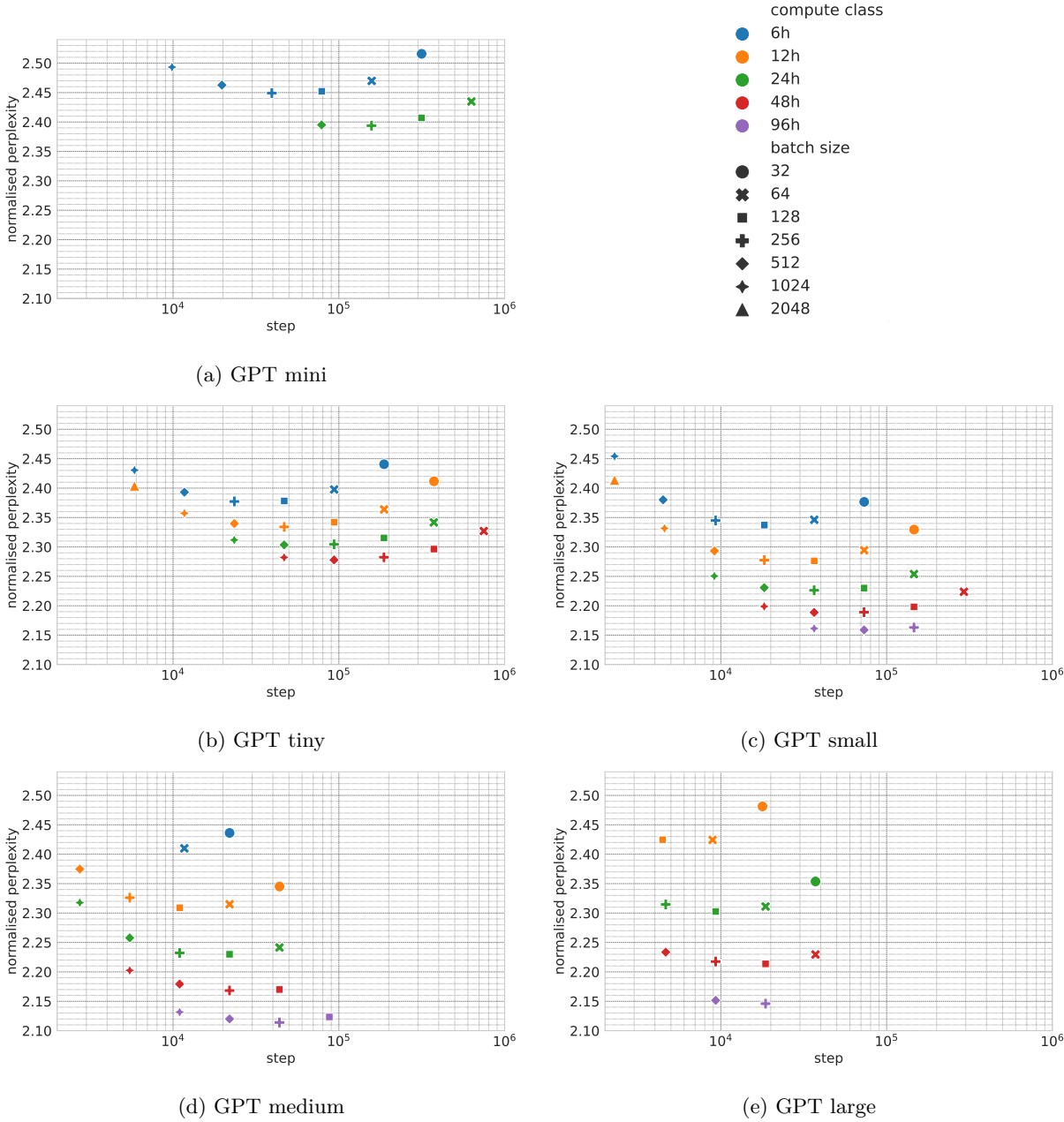

Figure 8: Fast evaluation of normalised perplexity on held-out data for GPT models of different sizes trained in different compute classes with different trade-offs between batch size and number of training steps. All models seem to have a similar compute-optimal batch size despite large differences in model size.

## 4.3 The Recurrent Baseline

Recurrent Neural Networks (RNNs), particularly the Long Short-Term Memory (LSTM; Hochreiter & Schmidhuber (1997)), have been a cornerstone in the development and evolution of deep learning for sequential data. RNNs are universal function approximators, i.e. they can be seen as general computers with finite state that can process arbitrary long inputs with constant memory and compute which is linearly proportional to the length of the sequence (Siegelmann & Sontag, 1991; Siegelmann, 1996; Schmidhuber, 1990). In contrast, the transformer model from Section 4.2, while also Turing complete (Pérez et al., 2019), requires a quadratic increase in compute and memory, and, in practice, cannot effectively process arbitrary

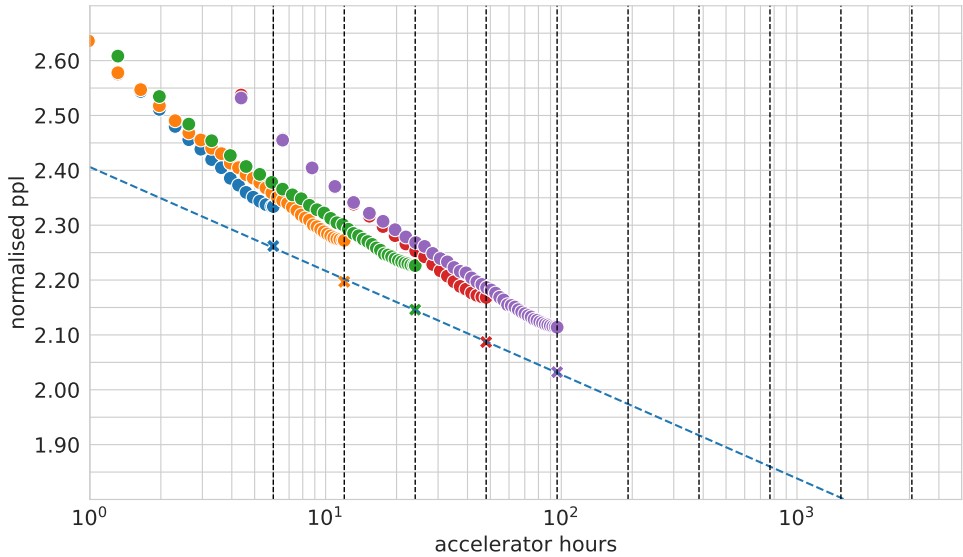

(a) Slope: −0.082 Intercept: 2.406

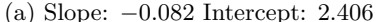

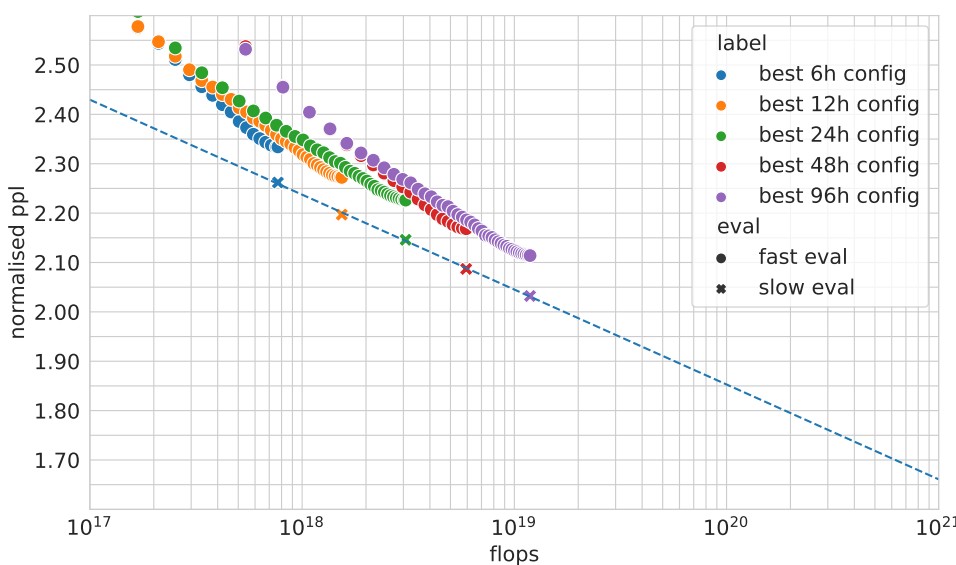

(b) Slope: −0.084 Intercept: 5.737

Figure 9: Scale plot of the best GPT configs evaluated on the test split. Fast eval considered predictions over tokens in the sequence. Slow eval evaluates only on the last 128 predictions which increases the minimum context per token from 1 to 384. Top: normalised perplexity over accelerator seconds. Vertical dashed lines are the different compute classes starting from 6h. Bottom: normalised perplexity over giga FLOPs. Like Kaplan et al. (2020), we estimate the FLOPs of the backward pass to be two times the forward pass.

long sequences. Furthermore, recurrence can be an advantageous bias for sequence models which enables them to generalise to out of distribution settings in more systematic ways (e.g. see Anil et al. (2022)).

On language modelling benchmarks, however, transformer models have outperformed RNNs. The inherent sequential computation in RNNs limits their parallel processing capabilities, something easily achieved by attention models. Unlike RNNs which compute a vector representation step by step, attention models do so

by attending to the entire sequence at once. This makes it more resource-intensive but allows for time-based parallel processing. Consequently, attention models can fully leverage the parallel processing capabilities of today's accelerators.

There has been previous work on enhancing the parallel processing abilities of RNNs. One intriguing direction is the quasi-RNN (qRNN, Bradbury et al. (2016)). Typically, a qRNN has a recurrent function that can also be processed in parallel. To make this possible, the recurrent weight matrix is often removed. In this section, we introduce the quasi-LSTM (qLSTM), a qRNN which achieves significantly higher throughputs while staying as true to the original LSTM as possible. While the presented qLSTM still lags behind our GPT baseline in throughput, we find that its compute-optimal batch size is significantly smaller, and it also achieves a larger gain in perplexity while processing the same number of tokens. In comparison with the GPT baseline, the data efficiency counterbalances its relatively reduced throughput, allowing for a measurable performance on the Languini Books benchmark.

### 4.3.1 The Model

The qLSTM is a variation of the LSTM which is why we will first describe the LSTM model. Our LSTM model uses the same architecture as the Transformer baseline from Section 4.2 where the only difference is the multi-head attention sublayer which we replace with a multi-head LSTM cell. Analogous to the multi-head attention, the multi-head LSTM cell splits the LSTM cell into multiple heads which perform the same operation in a lower dimensional space. The following equations describe the classic LSTM cell adapted for one head:

$$\boldsymbol{f}_t = \sigma(\boldsymbol{W}_f \boldsymbol{x}_t + \boldsymbol{U}_f \boldsymbol{h}_{t-1} + \boldsymbol{b}_f) \tag{11}$$

$$\boldsymbol{i}_t = \sigma(\boldsymbol{W}_i \boldsymbol{x}_t + \boldsymbol{U}_i \boldsymbol{h}_{t-1} + \boldsymbol{b}_i) \tag{12}$$

$$\boldsymbol{z}_t = \phi(\boldsymbol{W}_z \boldsymbol{x}_t + \boldsymbol{U}_z \boldsymbol{h}_{t-1} + \boldsymbol{b}_z) \tag{13}$$

$$\boldsymbol{o}_t = \sigma(\boldsymbol{W}_o \boldsymbol{x}_t + \boldsymbol{U}_o \boldsymbol{h}_{t-1} + \boldsymbol{b}_o) \tag{14}$$

$$\boldsymbol{c}_t = \boldsymbol{c}_{t-1} \odot \boldsymbol{f}_t + \boldsymbol{i}_t \odot \boldsymbol{z}_t \tag{15}$$

$$\boldsymbol{h}_t = \boldsymbol{o}_t \odot \phi(\boldsymbol{c}_t) \tag{16}$$

$$\boldsymbol{x}'_t = \boldsymbol{W}_h \boldsymbol{h}_t \tag{17}$$

where $\boldsymbol{W} \in \mathbb{R}^{d_{\text{head}} \times d_{\text{model}}}$ are the feed-forward weight matrices and $\boldsymbol{U} \in \mathbb{R}^{d_{\text{head}} \times d_{\text{head}}}$ are the recurrent weight matrices of this head, $\boldsymbol{b} \in \mathbb{R}^{d_{\text{head}}}$ are bias vectors, $\boldsymbol{x}_t \in \mathbb{R}^{d_{\text{model}}}$ is the hidden representation of step $t$ of the current layer, $\boldsymbol{h}_{t-1} \in \mathbb{R}^{d_{\text{head}}}$ is the state representation of this head for step $t-1$, $\sigma$ is the sigmoid function, $\phi$ is the tanh function, $\odot$ is the Hadaramard product or element-wise multiplication of two tensors, $\boldsymbol{c} \in \mathbb{R}^{d_{\text{head}}}$ is the cell state of this head, $\boldsymbol{W}_h \in \mathbb{R}^{d_{\text{model}} \times d_{\text{head}}}$ is the projection back to the embedding size $d_{\text{model}}$, and $\boldsymbol{x}'_t$ is the output of the LSTM sublayer.

Despite the increased parallel structure of the muli-head LSTM, each head performs multiple small matrix multiplications for every step in the sequence. With large backpropagation spans, such as the 512 steps we do in all our experiments, this results in significant sequential computation and drops in the utilisation of the accelerator hardware. By dropping the recurrent weights $\boldsymbol{U}$ and the dependency of the gates on the previous state $\boldsymbol{h}_{t-1}$ we further increase the parallelisability and arrive at our multi-head qLSTM formulation:

$$\boldsymbol{f}_t = \sigma(\boldsymbol{W}_f \boldsymbol{x}_t + \boldsymbol{b}_f) \tag{18}$$

$$\boldsymbol{i}_t = \sigma(\boldsymbol{W}_i \boldsymbol{x}_t + \boldsymbol{b}_i) \tag{19}$$

$$\boldsymbol{z}_t = \phi(\boldsymbol{W}_z \boldsymbol{x}_t + \boldsymbol{b}_z) \tag{20}$$

$$\boldsymbol{o}_t = \sigma(\boldsymbol{W}_o \boldsymbol{x}_t + \boldsymbol{b}_o) \tag{21}$$

$$\boldsymbol{c}_t = \boldsymbol{c}_{t-1} \odot \boldsymbol{f}_t + \boldsymbol{i}_t \odot \boldsymbol{z}_t \tag{22}$$

$$\boldsymbol{h}_t = \boldsymbol{o}_t \odot \phi(\boldsymbol{c}_t) \tag{23}$$

$$\boldsymbol{x}'_t = \boldsymbol{W}_h \boldsymbol{h}_t \tag{24}$$

Note that the only sequential operation that remains is an element-wise linear map:

$$c_t = c_{t-1} \odot f_t + u_t \tag{25}$$

where we summarised $i_t \odot z_t$ into the update vector $u_t \in \mathbb{R}^{d_{\text{head}}}$.

**A parallel implementation of recurrence.** The sequential part of the qLSTM in Eq. 25 can be expanded over 4 steps of the sequence as follows.

$$c_t = c_{t-1} \odot f_t + u_t \tag{26}$$
$$c_t = (c_{t-2} \odot f_{t-1} + u_{t-1}) \odot f_t + u_t \tag{27}$$
$$c_t = ((c_{t-3} \odot f_{t-2} + u_{t-2}) \odot f_{t-1} + u_{t-1}) \odot f_t + u_t \tag{28}$$
$$c_t = (((c_{t-4} \odot f_{t-3} + u_{t-3}) \odot f_{t-1} + u_{t-1}) \odot f_{t-1} + u_{t-1}) \odot f_t + u_t \tag{29}$$

$$\begin{aligned}
c_t = \quad & c_{t-4} \odot f_{t-3} \odot f_{t-2} \odot f_{t-1} \odot f_t \\
& + u_{t-3} \odot f_{t-2} \odot f_{t-1} \odot f_t \\
& + u_{t-2} \odot f_{t-1} \odot f_t \\
& + u_{t-1} \odot f_t \\
& + u_t
\end{aligned} \tag{30}$$

We can rewrite Eq. 30:

$$[c_t]_j = \sum_i \begin{bmatrix} c_{t-4} \\ u_{t-3} \\ u_{t-2} \\ u_{t-1} \\ u_t \end{bmatrix}_{i,j} \begin{bmatrix} f_{t-3} \odot f_{t-2} \odot f_{t-1} \odot f_t \\ f_{t-2} \odot f_{t-1} \odot f_t \\ f_{t-1} \odot f_t \\ f_t \\ 1 \end{bmatrix}_{i,j} \tag{31}$$

$$\tag{32}$$

Or more generally, we can describe a tensor $\mathbf{F} \in \mathbb{R}^{4 \times 5 \times d_{\text{head}}}$ which consists of the following matrix of vectors

$$\mathbf{F} = \begin{bmatrix} f_{t-3} & 1 & 0 & 0 & 0 \\ f_{t-3} \odot f_{t-2} & f_{t-2} & 1 & 0 & 0 \\ f_{t-3} \odot f_{t-2} \odot f_{t-1} & f_{t-2} \odot f_{t-1} & f_{t-1} & 1 & 0 \\ f_{t-3} \odot f_{t-2} \odot f_{t-1} \odot f_t & f_{t-2} \odot f_{t-1} \odot f_t & f_{t-1} \odot f_t & f_t & 1 \end{bmatrix} \tag{33}$$

such that

$$\begin{bmatrix} c_{t-3} \\ c_{t-2} \\ c_{t-1} \\ c_t \end{bmatrix}_{k,j} = \sum_i \begin{bmatrix} c_{t-4} \\ u_{t-3} \\ u_{t-2} \\ u_{t-1} \\ u_t \end{bmatrix}_{i,j} \mathbf{F}_{k,i,j} \tag{34}$$

which allows for the parallel computation of all cell states. Given all cell states, we can then compute the sublayer outputs for each step simultaneously using Eq. 23 and 24.

Central to this approach is the tensor $\mathbf{F}$. In theory, $\mathbf{F}$, or parts of it, can be computed efficiently in linear time w.r.t. the sequence length using a parallel prefix sum algorithm. Although such an algorithm exists in the CUDA framework, it is currently not available in PyTorch.

| Model | $n_{\text{heads}}$ | $d_{\text{head}}$ | block length | tokens per second | implementation |
|---|---|---|---|---|---|
| LSTM small | 1 | 768 | - | 1,462 | minimal for-loop |
| LSTM small | 12 | 64 | - | 1,464 | minimal for-loop |
| qLSTM small | 1 | 768 | - | 4,499 | minimal for-loop |
| qLSTM small | 12 | 64 | - | 4,494 | minimal for-loop |
| qLSTM small | 1 | 768 | 1 | 2,235 | block-parallel |
| qLSTM small | 12 | 64 | 1 | 2,352 | block-parallel |
| qLSTM small | 12 | 64 | 16 | 11,143 | block-parallel |
| qLSTM small | 12 | 64 | 32 | 8,638 | block-parallel |

Table 7: The best throughputs achieved on the RTX 3090 for the LSTM and quasi-LSTM with different implementations, block lengths and number of heads. The minimal for-loop implementation computes Eq. 25 in sequence whereas the block-parallel implementation uses Eq. 34 to compute all tokens within a block in parallel. Increasing the block length results in higher throughput and higher hardware utilisation but is less memory efficient.

Nevertheless, we observe that computing **F** in native PyTorch for a few tokens can significantly improve the model's hardware utilisation and throughput. We refer to the tokens that we process in parallel as *blocks* and find that certain block lengths dramatically increase the throughput of our qLSTM models.

Running several experiments at different scales with models that achieve 100-fold less throughput is unfeasible. For this reason, we limit our experimental evaluation to the fastest LSTM variant, the Multi-Head qLSTM model with a block length of 8 or 16. For easier comparison, we define the qLSTM model sizes in Table 8 to be within roughly 10% of the parameter and/or total FLOP count of the respective GPT model sizes from Table 4.

| qLSTM Model | gigaFLOPs | Params | $d_{\text{model}}$ | $n_{\text{layers}}$ | $n_{\text{heads}}$ | $d_{\text{head}}$ | block length |
|---|---|---|---|---|---|---|---|
| mini | 19.9 | 27.8M | 512 | 4 | 8 | 32 | 16 |
| tiny | 44.3 | 55.9M | 768 | 4 | 12 | 64 | 8 |
| small | 107.3 | 117.3M | 768 | 12 | 12 | 64 | 16 |
| medium | 352.7 | 361.1M | 1024 | 12 | 16 | 64 | 16 |
| large | 780.4 | 787.0M | 1536 | 24 | 16 | 96 | 16 |
| XL | 1,633.6 | 1,628.2M | 2048 | 24 | 24 | 128 | 16 |

Table 8: Overview of our qLSTM model sizes, flops, parameters, and differing hyperparameters. FLOPs are the total number of floating point operations during the forward pass with a batch size of 1 as measured by the DeepSpeed flops profiler.

| qLSTM Model | RTX 3090 | RTX 4090 | V100-32GB | A100-80GB |
|---|---|---|---|---|
| mini | 94,781 (82) | 166,178 (82) | 82,318 (112) | 186,145 (284) |
| tiny | 36,930 (60) | 57,454 (61) | 33,533 (84) | 62,031 (213) |
| small | 11,143 (17) | 16,312 (16) | 9,060 (22) | 23,433 (63) |
| medium | 2,509 (5) | 2,601 (5) | 1,929 (7) | 6,731 (22) |
| large | 1,006 (2) | 1,002 (2) | 773 (3) | 4,064 (13) |
| XL | OOM | OOM | OOM | 1,720 (5) |

Table 9: Overview of the best tokens per second measures and their respective batch sizes in brackets for different qLSTM model sizes on several accelerators. OOM stands for out of memory.

### 4.3.2 Results

We present the best normalised perplexity scores for the compute classes from 6h to 96h in Table 10. In Figure 10b we compare the total FLOPs of the best qLSTM and GPT models. We find that the qLSTM models counter-balance their 5-6k times slower throughput with a faster convergence which requires fewer

tokens than the GPT models to achieve a similar perplexity. As a result, our qLSTM model beats the GPT baseline after roughly 2 exaFLOPs.

As indicated in the previous Section, our qLSTM implementation does not make ideal use of the accelerator hardware. In Figure 9a we compare our best qLSTM models with our best GPT models on the basis of accelerator hours and we find that the qLSTM lags behind on all scales up to 96h. However, we observe that the qLSTM achieves a steeper scaling law than the GPT models, indicating a cross-over point at roughly 50,000 accelerator hours.

| compute class | normalised perplexity | config | batch size | total train tokens | total exaFLOPs |
|---|---|---|---|---|---|
| 6h | 2.518 | mini | 80 | 2.05B | 0.239 |
| 12h | 2.463 | tiny | 80 | 1.60B | 0.414 |
| 24h | 2.361 | small | 84 | 0.96B | 0.605 |
| 48h | 2.280 | small | 80 | 1.93B | 1.211 |
| 96h | 2.215 | small | 160 | 3.85B | 2.421 |

Table 10: Normalised perplexity values of the best qLSTM runs for each compute class. The number of tokens and exaFLOPs are computed as in Table 6.

We compare the qLSTM with the GPT model on our out of distribution splits in Table 11 (and Figure 11 in the appendix). On the langlearn, discworld, and wood splits the models tend to have higher normalised perplexity. On the java and stats this is less the case. In fact, both 96h models have lower normalised perplexity on java and stats books than on the regular test split.

| compute class | model | normalised perplexity on ood splits | | | | |
|---|---|---|---|---|---|---|
| | | langlearn | discworld | java | stats | wood |
| 6h | GPT | 4.042 | 2.393 | 2.264 | 2.131 | 2.370 |
| | qLSTM | 5.168 | 2.670 | 2.772 | 2.531 | 2.719 |
| 12h | GPT | 3.744 | 2.326 | 2.165 | 2.062 | 2.293 |
| | qLSTM | 4.865 | 2.619 | 2.736 | 2.466 | 2.639 |
| 24h | GPT | 3.521 | 2.273 | 2.104 | 2.011 | 2.232 |
| | qLSTM | 4.525 | 2.526 | 2.588 | 2.354 | 2.511 |
| 48h | GPT | 3.330 | 2.217 | 2.036 | 1.948 | 2.153 |
| | qLSTM | 4.158 | 2.440 | 2.464 | 2.252 | 2.424 |
| 96h | GPT | 3.135 | 2.158 | 1.977 | 1.898 | 2.088 |
| | qLSTM | 3.834 | 2.343 | 2.324 | 2.176 | 2.325 |

Table 11: Evaluation of the best GPT and qLSTM models on all out of distribution splits.

## 5 The Languini Codebase

The Languini Kitchen codebase is fundamentally a research-focused codebase, created with the intent of being easy to use, comprehensible, and sufficiently performant for our benchmark. One of our primary objectives is to provide researchers with an environment that enables them to draw meaningful and equitable comparisons with prior works. Furthermore, Languini also serves as a platform for identifying promising methods that have the potential to be scaled up.

The codebase supports data parallel experiments but not model parallel. Model parallelism is necessary to scale up experiments to very large models, typically with billions of parameters or more, by distributing the model across multiple devices. However, few researchers will have access to such expansive computational resources and are thus outside the motivation of Languini. Models trained in Languini ought to fit within the GPU memory of the chosen reference hardware.

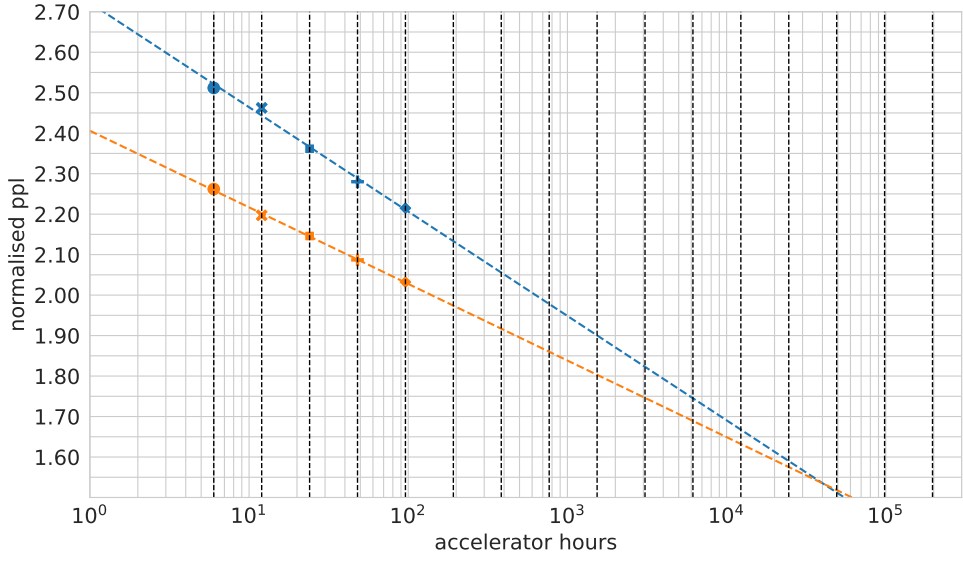

(a) qLSTM slope: −0.112 intercept: 2.722
GPT slope:−0.082 intercept: 2.406

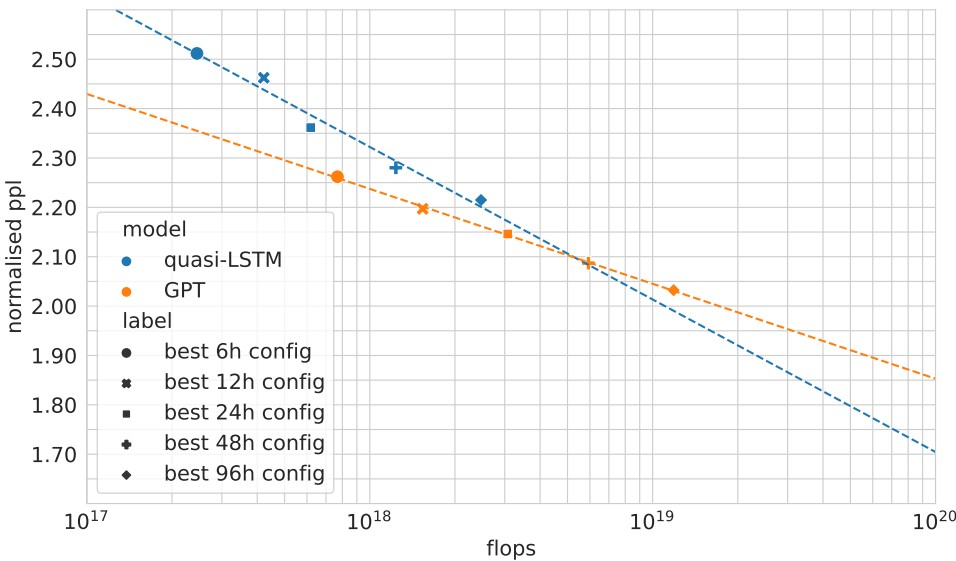

(b) qLSTM slope: −0.134 intercept: 7.883
GPT slope:−0.084 intercept: 5.737

Figure 10: Scale plot of the best qLSTM configs and the best results of the GPT models using the slow evaluation over the last 128 predictions per batch on the test split. Top: normalised perplexity over accelerator seconds. Vertical dashed lines are the different compute classes starting from 6h. Bottom: normalised perplexity over total FLOPs. Like previous work, we estimate the FLOPs of the backward pass to be two times the forward pass (Kaplan et al., 2020).

The Languini codebase is inspired by Scenic, a lightweight library for the development of vision models (Dehghani et al., 2022). It similarly provides various model-agnostic features, ranging from logging and data loading to training and evaluation functionalities. In order to maintain clarity and avoid complexity, experiments and the code will be placed in distinct and isolated project folders. Every research endeavour will

have its exclusive project directory, complete with its own necessary library code. This preserves simplicity and ensures that each project remains independent of subsequent advancements. For this reason, Languini will prevent interdependencies between projects. Once a project concludes, its respective folder ought to remain unchanged to guarantee reproducibility. Although this approach may lead to some code redundancy, we believe this is a justifiable trade-off for a research-based codebase to prevent the core from deteriorating.

To be listed in the Languini leaderboard, researchers are expected to provide not just the model code, but also configurations for all compute classes. Furthermore, every project has to provide scripts to download training logs and final model checkpoints from an archival hoster. We recommend utilising Zenodo.org, a reputable open repository developed by CERN under the European OpenAIRE program, for this purpose (European Organization For Nuclear Research & OpenAIRE, 2013).

The Languini Kitchen codebase is licensed under Apache 2.0, granting researchers the freedom to employ the code as they deem fit. Nonetheless, we urge researchers to contribute their most noteworthy results as a dedicated project folder, accompanied by instructions and a reference to their published work. This will further facilitate reproducibility and allow peers to draw comparisons with ease.

## 6 Open Research Questions

The field of language modelling has never been more exciting. With a benchmark where virtually all models are underfitting, it shifts the focus away from ad-hoc regularisation techniques to innovations that will hopefully be directly applicable at scale. In this section, we highlight just a few of the interesting directions that future work may explore.

**Better tokenisation.** Our exploration of common BPE tokenisation vocabularies, as detailed in Section 4.1, has brought several intriguing findings to light. Notably, many tokens can be derived using elementary symmetries. We also observed that the size of the vocabulary can substantially influence performance. These discoveries underscore the potential for innovative tokenisation methods. While recent studies underscore the benefits of byte-level modes, they remain inferior to BPE tokenisers in compute-constrained experiments.

**Implementational efficiency.** Recent work, such as flash attention, has highlighted the inefficiencies inherent in the native implementation of resource-intensive aspects of the model. Enhanced compilers, libraries, or a more in-depth understanding of a low-level implementation could boost the throughput of a model without necessitating conceptual changes. An example of such is Rockmate (Zhao et al., 2023b) which is a tool to make models more memory efficient at the cost of re-computing certain activations.

**Optimisation improvements.** While our experiments utilise simply Adam, there have been various advancements in the optimisation of language models. However, a recent study indicates that some perceived advantages diminish when experiments account for data or compute disparities (Kaddour et al., 2023). The Languini Books benchmark, being more expansive and akin to large-scale data than prior academic benchmarks, coupled with the existing model implementation within the Languini codebase, can facilitate a better assessment of novel optimisation techniques.

**Introduction of new models.** Languini provides a feed-forward and a recurrent baseline. Each approach has its unique strengths and limitations. Over the past few years, several models have been published which declare their supremacy over the decoder-only transformer model in some way, but few have demonstrated their scalability. Examples of such are the following: a Linear Transformer (Schmidhuber, 1991; Katharopoulos et al., 2020; Schlag et al., 2021) called TransNormer (Qin et al., 2023), a block-recurrent Transformer (Hutchins et al., 2022), novel parallelisable RNN called RWKV (Peng et al., 2023), or a state-space model for language modelling called H3 (Fu et al., 2023). Unfortunately, each one of them has been trained and evaluated on different data and hardware making a direct comparison impossible. The Languini Books benchmark, however, could serve as a platform for such models to demonstrate their benefits in a fair and reproducible way with scalability in mind.

**Advancement in theory.** The Languini Books benchmark boasts a significant enough scale to empirically demonstrate model-specific scaling laws. Furthermore, our preliminary results indicate that the compute-optimal batch size is also model-specific and depends weakly on the size of the model but more work is required to establish a principled approach that scales.

**Enhanced generalisation.** The Languini Books dataset incorporates several out of distribution splits. These splits mirror the divergence between the data on which the language model was trained and the context wherein it is deployed. The splits we introduced emphasize vast volumes of unique context that were removed from the training corpus, necessitating models to adapt and learn on the fly. Given the limited context of current models, this may demand novel strategies, possibly via efficient online learning algorithms or novel and dynamic architectures equipped with the capacity to meta-learn.

## 7 Conclusion

In this work, we introduced the Languini Kitchen, a research collective and codebase designed to democratize language modelling research by facilitating meaningful contributions across varying scales of computational resources. We presented an experimental protocol that emphasizes the use of accelerator hours as a more informative and equitable metric for comparison, addressing limitations inherent in the conventional measures of the number of parameters or FLOPs.

Utilising a filtered version of the books3 dataset, we demonstrated the utility of our approach in offering a fair and meaningful platform for comparing language models. We provided two baseline models, a feed-forward model based on the GPT-2 architecture and a recurrent model based on the new LSTM variation designed for larger throughput. Our empirical analysis revealed that while the GPT-2-based model performs strongly in absolute terms, the quasi-LSTM exhibits superior scaling laws, converging more efficiently with fewer tokens.

As a future direction, the scalability of our quasi-LSTM model offers intriguing possibilities for optimization and performance improvement. Furthermore, the Languini Kitchen's codebase is open for community contributions, encouraging ongoing research and development aimed at improving the performance of language models and identifying new candidates to be scaled up.

By setting new standards for fair comparison and offering tools for practical implementation, we believe that the Languini Kitchen lays the foundation for advancing the state of the art in language modelling research.

### Broader Impact Statement

The Languini Kitchen aims to democratize access to state-of-the-art language modelling research by creating an equitable framework for comparing performance across different scales of computational resources. In doing so, it opens up opportunities for researchers and institutions with limited computational capabilities to contribute meaningfully to the field. This democratization can lead to increased diversity in research perspectives, potentially yielding innovative solutions to existing problems and fostering greater inclusivity in the field of machine learning.

Lastly, it's worth considering that any advancements in language modelling, including those made more accessible by the Languini Kitchen, come with ethical implications related to data privacy, algorithmic bias, and the potential misuse of generated text. As the Languini Kitchen makes it easier to develop more capable language models, it also magnifies the importance of ensuring that these technologies are developed and deployed responsibly.

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

# A  OOD Scale Plots

The following are scale plots for the best GPT and qLSTM models on the out of distribution splits of the languini books dataset.

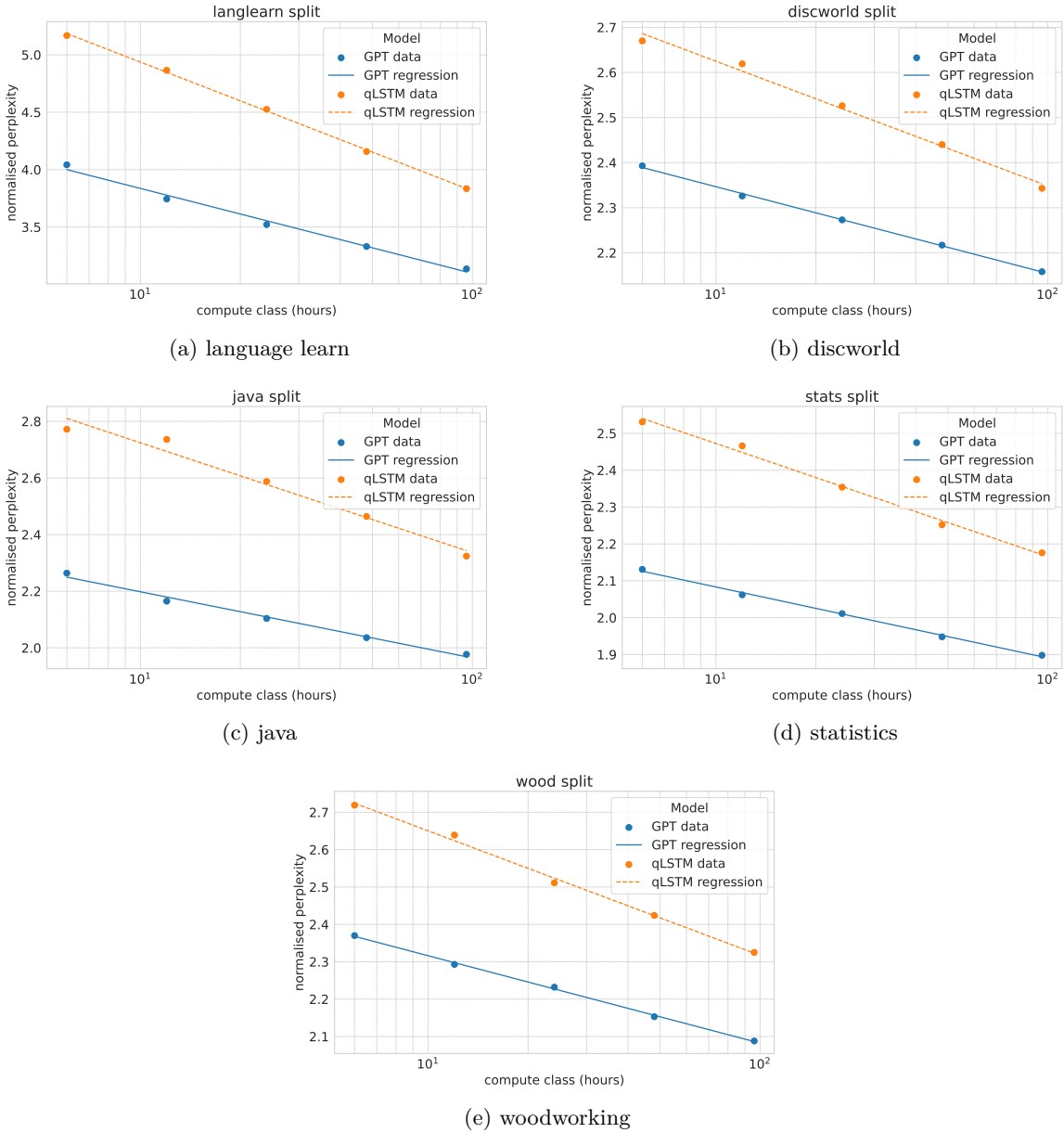

Figure 11: Scale plots on the ood data for the best GPT and qLSTM models.

