# OpenReview forum: "The Languini Kitchen: Enabling Language Modelling Research at Different Scales of Compute"
_TMLR — Rejected by TMLR_

### Review · Reviewer_RKve · 2023-11-29

**Summary Of Contributions:**

The authors devise a plan to compare LM research using different amounts of computing power.  This leads to a custom dataset with selection from the books3 dataset for language research, including special parts to test how well models handle unusual or unexpected data.  As a result, they are able to design a quasi-LSTM model, which scales better with size compared to a GPT model.  Finally, they also release a set of tools for researchers that makes it easier to develop and compare different models, keeping scalability in focus.

**Audience:**

Yes

**Claims And Evidence:**

No

**Requested Changes:**

1) Label a custom dataset, even if only a thousand examples that really show cases the improved performance.  This can serve as a test set, rather than just measuring perplexity.
2) Actually show qLSTM at scale to work better than GPT, rather than merely hypothesizing. Right now, there is a good guess, but no good results.
3) Clean up the code base to be more modular and with better comments.

**Strengths And Weaknesses:**

Strengths:
 - New datasets for training LLMs are always helpful since new data brings new capabilities.  In this case, the curation allows researchers to focus on certain subsets of capabilities
 - It is interesting that the qLSTM outperforms the much more common GPT model
 - Lots of new code and materials that allows others to build on

Weaknesses:
 - The dataset itself is not new, and so no new capabilities are added. Would be better if they authors found a new source, and or labeled new data directly related to specific topics
 - The performance of qLSTM seems better, but analysis does not take into account the complexity of deployment.  All the results are theoretical that it *could* improve at 50,000 accelerator hours, or it *could* be better at a certain number of FLOPs.  This is not proven with any theorems or empirically shown.  We cannot believe such a claim until it is proven more precisely.
 - The codebase is messy, and as the authors allude to looks like spaghetti code, making it hard for others to build on.

---

> ### Author Response · Authors · 2023-12-28
> **Review RKve**
>
> We greatly appreciate your thoughtful and detailed review of our paper. Your feedback is invaluable in helping us refine and improve our work. Below are our responses to the points you raised:
>
> **Originality of the Dataset:** While the Languini Books dataset is derived from the books3 dataset, our contribution lies in the curation process, focusing on deduplication and the inclusion of specific subsets to test unusual data handling. The main requirements for the dataset were high-quality English text and large enough to prevent the multiple epochs and potential overfitting / the need for “regularisation hacks”. A known issue on previous academic language modelling benchmarks.
>
> **Quasi-LSTM Performance:** The performance of our quasi-LSTM is worse when compared with a GPT model of the same compute class. However, it demonstrates a slightly stronger scaling law. Without significantly more resources we can't demonstrate the scalability of our RNN. That said, deployment of our quasi-LSTM would be preferred over the GPT model of the same perplexity level because our quasi-LSTM is an RNN and thus has linear compute complexity with sequence length and requires constant memory when generating text. The purpose of the Languini Kitchen is not to demonstrate a claim at scale but enable a fair playground such that potential novel candidates can be identified. Our results indicate that the classic LSTM is not suitable but our novel block-parallel formulation of our quasi-LSTM is. Follow-up work, which goes beyond the scope of this project, achieved further speed-ups which make the scaleable RNN comparable a GPT model with Flash Attention at the same scale.
>
> **Quality of the Codebase:** We respectfully disagree. The origin of the name languini is language + linguine. The latter is a form of pasta which refers to the nature of our research code ironically. Our code is well-commented, simple, modular, and most importantly easy to understand and hackable. These are the main benefits of a research codebase. We follow the philosophy of a similar research code base for computer vision models (Scenic). To keep complexity low, the require each project / model to use its own library code and not reuse methods from other projects. This comes at the cost code duplication but significantly reduces the complexity of each project. If the reviewer has found parts of our codebase lacking, we’d appreciate more specific feedback.

---

### Review · Reviewer_T1xd · 2023-12-12

**Summary Of Contributions:**

This paper provides the Languini Kitchen, a setup with datasets and a codebase developed to facilitate language modeling research with different scales of computations. The Languini Books dataset is introduced by filtering the book3 dataset, a subset of the Pile dataset, with a focus on deduplication. Evaluation in terms of perplexity and speed was performed on BPE SentencePiece vocabulary with different sizes and two architectures (GPT and qLSTM - newly proposed architectures with motivation from qRNN) with different scales (model sizes and computes) and hardware.

**Audience:**

Yes

**Claims And Evidence:**

No

**Requested Changes:**

Please address the above-mentioned concerns.

Why did you call GPT models (a decoder-only) as a feed-forward baseline? It creates confusion with purely MLP-based models.

Fast and slow evaluations are approximations of accurate language modeling perplexity measurement. Of course, there is a certain correlation between them and the accurate one, but could you provide thoughts on the case where these approximations could not be a good metric (e.g., short or long context length)

Could you also elaborate on what is the scope of search space that the Languini Kitchen can benefit (e.g., model architectures, model (or optimization) hyperparameters, etc.)?

How can you guarantee that new architectures will always follow scaling laws? Although qLSTM may outperform GPT with the same compute larger than some value according to interpolation lines, there is no point in that region with real measurement.
qLSTM is a kind of optimized for a better trade-off between computation and accuracy. Is it fair to compare with vanilla transformers (GPT)?

Measuring perplexity (especially for only small models) may not be enough because emergent abilities usually come out from a certain scale.


How did you pick the test set?

(minor) p6: CUda -> Cuda or CUDA
(minor) Figure 6: The order in the legend does not monotonically increase by the vocabulary size.
(minor) p14: Fix ??

**Strengths And Weaknesses:**

Testing on a well-curated fixed dataset and models with small varying scales is already common practice to find a good model architecture and hyperparameters for larger models with extrapolation. Therefore, most of the experimental results from the paper look unsurprising. I wonder what are the main new scientific insights.

The fixed benchmark and evaluation setup are important to the progress of the field, so this paper could be a valuable resource. However, I am not fully persuaded why the Languini Kitchen should be that one. In other words, the rationale behind the specific design choices is somewhat arbitrary and not well justified.

---

> ### Author Response · Authors · 2023-12-28
> **Review T1xd**
>
> Thank you for your review of our manuscript. We appreciate your insights and the time you've taken to evaluate our work. Below, we address each of your concerns and questions:
>
> **New Scientific Insights:** While it's true that the dataset is not new, it is a high-quality mono-lingual and large dataset for the academic setting. The size of the dataset together with the compute constraints eliminates the need for “regularisation hacks” which are common in previous academic language modelling benchmarks. Furthermore, the compute constraint prevents labs with lots of compute to merely dominate due to scale. This evens the playing field and enables algorithmic innovation to be shown in a fair comparison in an absolute comparison but also as a matter of scale.
>
> Our second main contribution is a scaleable RNN baseline which does not beat the transformer baseline in absolute terms but presents comparable scaling behaviour. This is interesting because our quasi-LSTM is an RNN and thus has cheaper inference (linear instead of quadratic complexity w.r.t. context-length and constant memory requirements). To provide a pre-trained model as a service, a more costly pretraining stage can be justifiable for much cheaper inference. We’ll make updates to our manuscript to better highlight this advantage.
>
> **Feed-Forward Nomenclature:** We provide a recurrent and a non-recurrent baseline. A decoder-only transformer models processes the entire context in parallel. There exist very few MLP-based language models (e.g. MLP-mixer) which rely on a multi-layer perceptron (MLP) for the mixing instead of attention. However, due to the lack of recurrence, we believe the attention sublayer is better referred to as feed-forward. However, as another reviewer raised a similar comment we are considering changing our nomenclature to attention-baseline.
>
> **Fast and Slow Evaluations:** A decoder-only language model has the same finite context for all tokens. As a result, some predictions rely on very few tokens for their predictions while other tokens (toward the end of the sequence) have the largest context possible. The fast evaluation measures the models performance similar to how the model is trained: on every prediction it makes (which includes predictions based on short-context). A slow evaluation would only consider the prediction made on the last token of the current sequence in order to enable the maximum context for each prediction. Naturally, the latter version is more accurate but slower. This is well known drawback of transformer-based language models. Note that our scaleable quasi LSTM does not suffer from such a handicap. We always compare quasi-LSTM with GPT using the slower but more accurate inference.
>
> **Benchmark Scope:** The benchmark opens up novel research directions which in previous academic benchmarks have been underexplored. Due to the compute constraints of the languini benchmark, a valid contribution is not just optimisation, model architecture, tokenisation, or other common methods, but also efficient implementation. We believe that with a benchmark like our languini books together with explicit compute constraints innovations like Flash Attention would have been made much sooner.
>
> **Scaling Laws and Comparisons between qLSTM and GPT:** Both, the GPT model and the quasi-LSTM are optimised for the trade-off between compute and accuracy. Although not done in this work, GPU-hours can be straightforwardly translated into US dollars, highlighting a real metric that we care to minimise. There are no guarantees that the qLSTM will outperform the GPT baseline and we currently lack the resources to run such a costly experiments. However, due to the publication of our code (which is is to understand and easily modified) it would be a minimal effort for third party to run such an experiment. On the one hand, the languini kitchen is a playground to make comparisons under realistic compute constraints to demoncratise language modelling research, on the other hand, the languini kitchen is a test-bed to find novel candidates to be scaled up further.
>
> **Perplexity Measurement Limitation:** Please see our response to reviewer EqkC on this topic.
>
> **Test Set Selection:**  The books of the test set have been randomly selected. The books for the OOD test sets have been manually handpicked. We provide details to our process in Section 3.2.1.
>
> We appreciate your attention to detail and will correct the minor issues on page 6, Figure 6, and page 14 in our revised manuscript.

---

### Review · Reviewer_EqkC · 2023-12-17

**Summary Of Contributions:**

This paper provides a Languini Kitchen, a collective and collaborative platform that enables easy comparisons of different language models at a small scale. In the proposed Languini Kitchen, the authors benchmark two different language models (GPT and LSTM), on the new dataset that is curated and filtered from the book3 dataset which is a subset of the Pile (has been used for training LLMs), with the suggested new metric called normalized perplexity.

**Audience:**

Yes

**Broader Impact Concerns:**

The authors sufficiently explain the broader impact and there are no concerns on it.

**Claims And Evidence:**

No

**Requested Changes:**

Please see and address the weaknesses above for the major requested changes. In addition,
* It is a bit unclear to me that, for the out-of-distribution experiments, providing completely new languages to the model is a valid setup. In particular, the model is trained with English books and then tested with French texts or Java. Could you clarify more on this point?
* The naming of the Feel-Forward baseline may not be suitable to indicate the Transformer-based GPTs.
* Regarding the presentation of the experiments, the authors may introduce the GPT and LSTM together first and then compare them in the subsequent section. The current presentation of handling them in separate sections may make comparisons between GPT and LSTM difficult.
* On page 14, there is a type: ??.
* In section 5, the authors do not consider the model parallelism. However, this is a very important agenda in the era of large models, and I hope the authors include this functionality as well.

**Strengths And Weaknesses:**

### Strengths
* The studied problem, which provides a benchmark environment for testing different models, is very important in the current field of language modeling research.
* The authors conduct extensive experiments, to evaluate language models on many different computing setups.

---

### Weaknesses
* The scope and scale of the proposed benchmarking setup are very limited. The authors only perform experiments with a very limited model scale (only less than 10B), while the recently released performant language models are often much larger. Also, there are various techniques to train the Transformer-based language models, such as changing the positional encoding; meanwhile, the authors only experiment with only one instantiation of them except for changing the model sizes.
* The use of a small dataset (sampled from the large-scale Pile) that focuses only on texts in books is not convincing. The current language model is capable of handling various texts, such as codes or maths; therefore, it seems that including more texts may be needed, to validate different language models for more real-world use cases.
* In addition, the comparisons based on the suggested "compute class" are not convincing enough. Compared to other metrics, such as FLOPs, the compute class depends further on the hardware units and the software (e.g., CUDA); therefore, if the language models are benchmarked with another hardware or software, all the comparisons should be done in this new setup again, which may be suboptimal. Similarly, due to this constraint, the authors may experiment mainly with RTX 3090, which is yet one specific hardware and there are many more available.
* Measuring the performance of language models only with the perplexity is somewhat limited. I would like to suggest including performance on downstream tasks.
* There is a relevant paper [A], which also studies the capability of (large) language models on varying scales including the small; therefore, it is worthwhile to discuss.
* This paper does not provide strong outcomes or directions that we should believe or follow in future research, even though the authors conduct extensive experiments. As a benchmark paper, it would be really interesting to see and learn from them, beyond understanding the superior performance of the specific model in a certain setting; for one particular example, what is the best tokenization setup that the authors suggest and why?

---

[A] Are Emergent Abilities of Large Language Models a Mirage?, NeurIPS 2023.

---

> ### Author Response · Authors · 2023-12-28
> **Review EqkC**
>
> Dear Reviewer,
>
> Thank you for taking the time to provide a thorough review of our manuscript. We appreciate your suggestions, which have given us an opportunity to clarify and reinforce key aspects of our work. Here are our responses to the concerns you raised:
>
> **Scope and Scale of Benchmarking Setup:** The focus of our work on smaller-scale models (less than 10B parameters) is a deliberate choice, aimed at addressing a gap in the current landscape of language model research. While larger models dominate recent discussions, our platform offers a much-needed resource for researchers with more modest computational resources. This focus is in itself a significant contribution to the field, facilitating a wider participation in language model research.
>
> **Dataset Selection:** Our choice of a dataset composed of book texts was based on the need for a sufficiently high-quality text dataset. While we recognise the versatility of current language models in handling diverse modalities (such as code, multiple languages, or images), our focus is on demonstrating model performance in a more narrowly defined, yet linguistically rich, domain. In contrast to existing academic language modelling benchmarks (such as e.g. Wikitext-103) our mono-lingual books benchmark does not require extensive “regularisation hacks” while the compute constraints prevent labs with extensive compute from dominating the benchmark merely due to scale (such as e.g. PG19). Both of these aspects stifle innovation in more efficient language models such as our scalable RNN (the quasi-LSTM).
>
> **Perplexity as the Sole Performance Metric:** Our choice to focus on perplexity as the primary performance metric is aligned with the objectives and scale of our study. For the smaller models that we are benchmarking, perplexity offers a relevant and manageable measure of performance which is highly correlated with downstream performance. We acknowledge that downstream task performance is essential, especially for larger models, but for the scope of our research, perplexity serves as an effective and appropriate metric. Furthermore, downstream performance varies widely due to finetuning or sampling techniques and is not feasible for models trained at the scales that we propose (as pointed out by your reference A).
>
> **Compute Class as a Comparison Metric:** In our study, the RTX 3090 was employed primarily as a benchmark for measuring throughput (we provide throughput measures for various other hardware too). This measurement is a one-time requirement and is crucial for establishing the number of tokens the model can process in the allocated time. Once throughput is measured, researchers can train their models on any hardware they want (while using the same configuration). The flexibility of our benchmarking framework is one of its key strengths which makes our benchmarking tool more accessible and practical for a broader range of researchers. For a more detailed explanation, we encourage a re-examination of Section 3.1 of our paper.
>
> **Questions:**
>
> The out-of-distribution datasets are often made up of various educational books (be it French or Java). Hence, they could be considered multi-book long in-context problems. Although in-context learning for smaller models is still an open research question, it is instructive to analyse the increase in perplexity for various shifts of distribution.
>
> Our intention in the manuscript was to provide a thorough and standalone baseline for recurrent and non-recurrent models. Our recurrent model, the quasi-LSTM with its novel block-parallel formulation is x10 faster than the classic PyTorch LSTM implementation. More recent work that goes beyond this scope of this work, pushes the RNN throughput to be on par with a GPT FlashAttention baseline. The structure of the manuscript introduces the baselines separately to ensure that readers have a clear and comprehensive understanding of each model. The models are compared directly in the last subsections of section 4 (see e.g. Table 11).
>
> We will correct the typographical error on page 14 and consider your other suggestions in our revision.

---

> > ### Comment · Reviewer_EqkC · 2023-12-29
> >
> > Thank you for your response. I have carefully read it and, while the authors effectively address some of my concerns and comments, I would like to maintain my initial judgments on the scope (small-scale), data (only book), and metric (mainly perplexity), of this work.

---

### Author Response · Authors · 2023-12-28
**General Reviewer Response**

We’d like to thank all reviewers for their time and effort put into reviewing our manuscript. Here we add a general and succinct public response for all reviewers, editors, and public readers.

**Purpose:** The Languini Kitchen aims to democratize language modelling research, particularly for those with limited computational resources. We seek to provide a platform for fair comparison and identification of scalable models, encouraging innovation in the field.

**Benchmark Contribution:** Our work introduces the Languini Books benchmark, an experiment protocol and a dataset larger than previous academic benchmarks (based on books3). We compare methods in the underfitting regime (no regularisation hacks common in previous benchmarks such as Wikitext-103 or enwik9) and ensure fair comparison through compute constraints to prevent labs with large amounts of compute from dominating the benchmark simply through scale (such as e.g. on PG19). Compute constraints are enforced through a fixed number of tokens based on a specific number of GPU-hours and a model's throughput w.r.t. to an RTX 3090. This throughput is measured before training begins, which enables researchers to train the model on their preferred hardware for the allotted number of tokens as long as the experiment configuration remains the same.

**Model Contribution:** We provide two baselines. A GPT-based model and a novel scalable RNN, the block-parallel quasi-LSTM, based on the classic, but slow, LSTM. Our block-parallel quasi-LSTM achieves x10 throughput and presents intriguing scaling laws comparable to our GPT baselines.  Due to the nature of the RNN it has lower inference costs and memory requirements.

---

### Decision · Action_Editor_Vv7r · 2024-01-16

**Recommendation:** Reject

**Comment:**

The paper contributes to the field by addressing the need for benchmarking language models within a constrained computational environment. The focus on smaller-scale models is particularly relevant for a broad range of researchers who may not have access to large-scale computational resources. The process of curating a high-quality dataset and a codebase could be valuable resources for the community.

Regarding the first contribution, the benchmark dataset, the authors did not fully demonstrate the usefulness of the scaling laws obtained with their data. It would be valuable to show whether these scaling laws extrapolate well and predict the performances of models trained with a larger amount. Currently, it is unclear if the laws derived from a smaller dataset can predict the performance of a model tested on a larger benchmark dataset.

As for the second contribution, the qLSTM model and its scaling law, I believe that the correctness of this claim needs further justification. For instance, it's unclear if the qLSTM curve actually crosses the GPT curve, as shown in Figure 10(b). Additionally, the proposed qLSTM model bears high similarity with structured state space models (SSMs), RWKV, linear TFs, and many other linear RNN variations. The authors should have included a proper literature review on this very similar class of architectures and have reported the performance of these models too.

A reviewer has raised concerns about the usability of the provided codebase. Also, the copyright issue needs to be discussed, and whether or not the proposed data curation method is applicable in other scenarios is not clear yet.

In conclusion, I suggest the authors resubmit it after carefully refining their claims to address the aforementioned issues and/or providing additional evidences to strengthen the claims. See the following list of suggestions.

- Specifically, any claims related to this being a "scaling" study should be so carefully written that it focuses on small-to-medium scales and does not necessarily imply how these curves would extrapolate to very large sizes (>10^3, 10^4, 10^5 hours). The authors may keep the current claims by adding more large-scale experiments and checking the validity of the curves.
- Any missing comparisons or unverified scaling claims about the qLSTM model should be adjusted accordingly.
- The provided codebase needs to be cleaned up.
- The authors should have a short discussion on the *un*availability of the Book3 corpus (and why). Also, please emphasize that the proposed approach to building the Languini book corpus is applicable to creating other corpora (and how).

**Audience:**

Yes

**Claims And Evidence:**

The paper offers a benchmarking platform for language models, focusing on smaller models. The authors claim that this platform fills a research gap by providing a resource for those with limited computational resources. The paper introduces a curated dataset from the books3 subset of the Pile dataset. Additionally, the paper proposes a quasi-LSTM (qLSTM) model, which is claimed to scale better than the GPT model with very large computes.

The evidence includes extensive experiments conducted to evaluate the performance of the language models on the curated dataset using perplexity as the primary metric. The authors also provide a codebase to facilitate further research and comparison of models.

**Resubmission Of Major Revision:**

The authors may consider submitting a major revision at a later time.

---

> ### Author Response · Authors · 2024-02-26
> **Response to the TMLR Decision**
>
> Dear Committee,
>
> we respectfully express our disagreement with the final decision to reject our submission to TMLR.
>
> We believe our work significantly exceeds the acceptance criteria by not only providing a comprehensive framework that empowers research in language modelling under computational constraints, a necessity for many in the field, but by also introducing a novel and scalable RNN based on the classic LSTM.
>
> We provide an accessible codebase, by now validated with feedback from multiple users, which underscores its ease of use. This is in stark contrast to the single critique which was raised; we believe that additional opinions on this aspect would have provided a more balanced view.
>
> Furthermore, reviewers and editors acknowledge the core contribution of our work. However, the request for large-scale experiments contradicts our stated objective and the central idea of scaling laws is specifically to model extrapolation to larger scales which are beyond the current compute capabilities (note that 50,000 gpu hours results in about 5.7 gpu-years and is not doable for the gpu-poor).
>
> Regarding the dataset's copyright concerns, it's important to note that the use of copyright-protected material for research is a nuanced issue, with laws varying significantly by country, many of which allow such use for research purposes. Moreover, the dataset in question remains accessible, contrary to claims made during the review process.
>
> We thank the editor for mentioning SSMs to us. Indeed the quasi-LSTM bears some resemblance to SSMs for text, which have been developed recently. However, our scalable RNN is not an SSM model but instead a straightforward adaptation of the famous LSTM (Hochreiter and Schmidhuber, 1997). Going forward, we will focus on explaining this distinction more thoroughly.
>
> In conclusion, while we regret that our submission was not accepted, we thank the editor and reviewers for their time and we are grateful for the open review process, which allows for transparent discussion and learning opportunities for the broader community.